# LogicVista: Multimodal LLM Logical Reasoning Benchmark in Visual Contexts

## Abstract

We propose LogicVista, an evaluation benchmark that examines multimodal large language models' (MLLMs) integrated **Logic**al reasoning capacities in **Vis**ual contexts. Recent advancements in MLLMs have demonstrated various fascinating abilities such as crafting poetry based on an image to engaging in mathematical reasoning. Despite these feats, there remains a gap in the systematic examination of MLLMs' proficiency in logical reasoning tasks. These skills are routinely invoked in navigation, puzzle-solving, etc. Thus we present LogicVista, which evaluates general logical cognition abilities across a spectrum of 5 logical reasoning tasks with 3 broad capabilities and 11 specific capabilities through a sample of 448 multiple-choice questions. Each is annotated with not only the correct answer but also the human-written reasoning behind the selection, allowing for rich open-ended evaluation as well as MCQ evaluation. A total of 11 MLLMs undergo comprehensive evaluation using LogicVista. We are also introducing a crowdsourced annotation tool to further scale LogicVista with support from the community. Code and Data Available at `https://anonymous.4open.science/r/LogicVista`.

## 1 Introduction

Recent advancements in Large Language Models (LLMs) are gradually turning the vision of a generalist AI agent into reality. These models exhibit near-human expert-level performance across a variety of tasks and have recently been augmented with visual understanding capabilities, enabling them to tackle even more complex visual challenges. This branch of work, led by both proprietary projects such as Flamingo (Alayrac et al., 2022) and GPT-4 (OpenAI et al., 2024) and open-source works such as LLaVA (Liu et al., 2023a), Mini-GPT4 (Zhu et al., 2023) enhances existing LLMs by incorporating visual comprehension. These models, known as Multimodal Large Language Models (MLLMs), utilize LLMs as the foundation for processing information and generating reasoned outcomes (Yin et al., 2023), bridging the gap between language and vision. Recent MLLMs have demonstrated a range of such intriguing abilities, such as writing poems based on an image (Fu et al., 2023a), engaging in mathematical reasoning (Alayrac et al., 2022), and even aiding in medical diagnosis (Zhang et al., 2023).

**Challenges** Many works have benchmarked MLLM's performance on common multimodal tasks such as recognizing objects (Antol et al., 2015), understanding the text in an image (Singh et al., 2019a), or performing math (Yu et al., 2023; Lu et al., 2024). However, there are two major concerns with existing benchmarks: lack of evaluation of explicit logical-visual-language reasoning skills and potential data leakage in benchmarking data.

Evaluating explicit logical-visual-language reasoning is essential, as it reflects a key aspect of human creativity and intelligence. Proficiency in reasoning skills is widely recognized as a reliable indicator of cognitive ability across various domains (Kahneman, 2012; Träff et al., 2019). While many datasets and benchmarks have been designed to assess the logical reasoning capabilities of AI agents, most are limited to text-based formats, leaving visual reasoning largely underexplored (Liu et al., 2023; Xu et al., 2023; Yang et al., 2023; Lin, 2024; Yang et al., 2024). While some datasets like GLoRE, MathVista, MM-vet, and RAVEN (liu et al., 2023; Lu et al., 2024; Yu et al., 2023; Zhang et al., 2019) have explored aspects of visual logical reasoning, they focus primarily on specific tasks such as mathematical reasoning, spatial reasoning, or world knowledge retrieval, with logical reasoning only partially embedded and not directly analyzed. General-purpose visual question answering and

captioning datasets like TextVQA and VQAv2 (Goyal et al., 2017a; Singh et al., 2019a) contain even fewer examples of visual logical reasoning, concentrating instead on the recognition and identification of visual details. Similarly, specialized benchmarks such as MMMU and OlympiadBench (Yue et al., 2024; He et al., 2024) focus on academic domain questions in subjects like math, science, or history, without directly evaluating the visual logical reasoning capabilities of modern MLLMs.

Moreover, many existing benchmarks rely on publicly available data from the internet, which can easily be included in the training datasets of various models due to its low friction for scraping (as demonstrated in Appendix K). This increases the likelihood that many benchmarking samples are inadvertently leaked into the training data, leading to unfair comparisons of models that do not effectively isolate their reasoning capabilities. In Appendix A, we provide a more comprehensive overview of the gaps in the current literature on MLLM benchmarks.

Recently, MLLMs have demonstrated impressive problem-solving and understanding capabilities across various domains. Researchers have aimed to strengthen these models' logical reasoning abilities through novel pre-training techniques, such as directly embedding logical reasoning, as demonstrated with IDOL (Xu et al., 2023). However, their capacity for explicit visual logical reasoning remains largely untested in a comprehensive, systematic way. Thus, developing a scalable and thorough benchmark to assess MLLMs' visual logical reasoning abilities is essential. This would drive advancements in logical reasoning systems within visual contexts, especially as VQA agents gain traction in fields like robotics, biology, and software engineering (Muennighoff et al., 2024; Hong et al., 2023; Xiao et al., 2024), while also providing a framework to evaluate progress in visual understanding and reasoning in MLLMs.

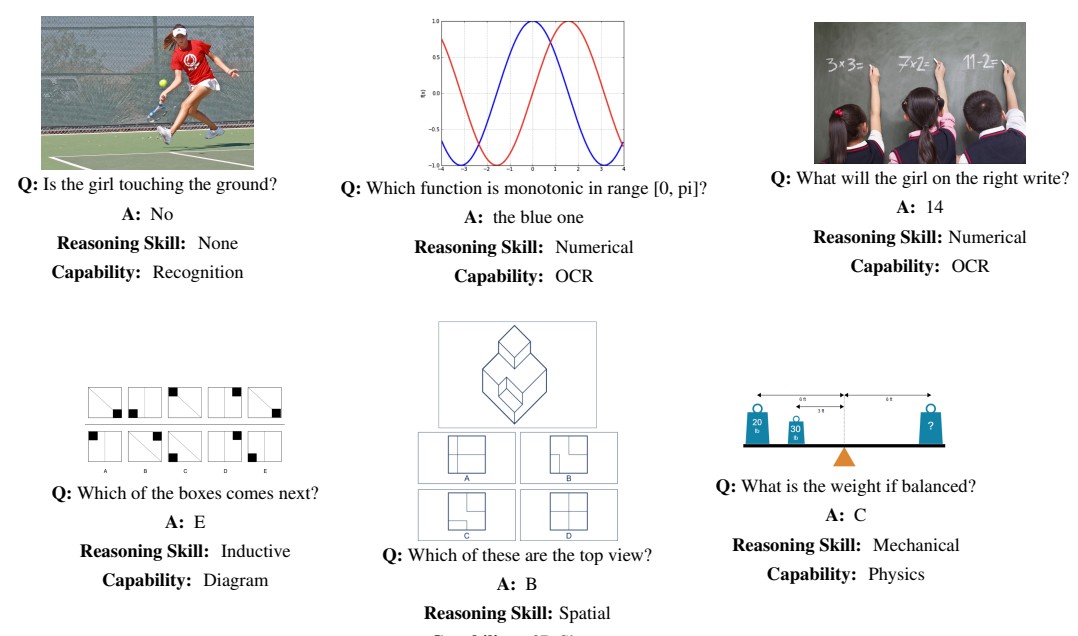

Figure 1: Capabilities and reasoning skills of different existing benchmarks. The top row shows examples from **VQAv2, MathVista, and MM-vet** in order from left to right, while the bottom row contains examples from our **LogicVista**. Unlike previous benchmarks, LogicVista focuses on visual reasoning capacities explicitly.

**This Work** With these motivations, we propose a comprehensive benchmark for general visual logical reasoning to address these challenges. Our benchmark utilizes rigorously sourced data to ensure high quality and fair evaluation of the explicit visual-logical reasoning skills of current state-of-the-art MLLMs. We argue that an effective universal evaluation benchmark should have the following characteristics: (1) coverage of a broad range of human logical reasoning skills, including deductive, inductive, numeric, spatial, and mechanical reasoning; (2) presentation of information in various formats such as Optical Character Recognition (OCR), graphs, charts, and flow diagrams to accommodate diverse data inputs; (3) responses structured for convenient quantitative

analysis, enabling rigorous assessment and comparison of model performance; and (4) scalability to accommodate community feedback and growth, ensuring sustainability and effective evaluation of the benchmark for future models and formats.

To this end, we collect a comprehensive MLLM evaluation benchmark, named LogicVista, which fulfills all the criteria:

- LogicVista covers the examination of 5 representative categories of logical reasoning tasks: inductive ($sample = 107$), deductive ($sample = 93$), numerical ($sample = 95$), spatial ($sample = 79$), and mechanical ($sample = 74$).
- LogicVista covers 3 broad capabilities and 11 specific capabilities to give a comprehensive view of how well MLLMs reason with various visual formats.
- All images- instructions-solution-reasoning are rigorously manually annotated and validated using our robust annotation pipeline.
- Benefiting from our instruction design "please select from A, B, C, D, and E." and our LLM answer evaluator, we can evaluate different reasoning skills and capabilities and easily perform quantitative statistical analysis based on the natural language output of MLLMs. We also provide more in-depth human-written explanations for why each answer is correct for more through open-ended evaluation.
- To ensure the scalability and sustainability of LogicVista for future evaluations, we introduce the annotation tool used for community crowdsourcing, as detailed in Appendix L.

As shown in Figure 3, LogicVista covers a broad range of reasoning skills, evaluating both open- and closed-source SOTA MLLMs. For example, the question *"Which of these images is the top view of the given object"* in Figure 1(b) requires spatial reasoning from a different perspective, not just object recognition. Since these questions and diagrams are presented without real-world context, they test the MLLM's core reasoning abilities.

We perform comprehensive evaluations on 11 representative open- and closed-source MLLMs, using 448 samples across 5 key logical reasoning categories, providing the first in-depth assessment of visual logical reasoning in state-of-the-art models like GPT-4 Omni, Claude-3.5 Sonnet, and Gemini-Pro. LogicVista's evaluation framework breaks down each model's performance by reasoning skill and capability, offering more nuanced insights than a single overall score. We employ two evaluation methods: MCQ for quick assessments and open-ended chain-of-thought (CoT) for a deeper analysis of the reasoning process, identifying where models succeed or fall short. This approach offers a clearer understanding of each model's strengths and weaknesses.

Our findings indicate that LogicVista is a highly challenging benchmark, with top-performing models averaging 65% in deductive reasoning but scoring below 30% in other reasoning categories. Notably, GPT-4o and Claude 3.5 Sonnet exhibit state-of-the-art performance on LogicVista, as detailed in Table 2. We observe that most models struggle the most with inductive, numerical, and spatial reasoning, while performing better in deductive and mechanical visual reasoning tasks. Additionally, our analysis shows that MLLMs achieve higher accuracy with MCQ prompts compared to CoT-based prompts, suggesting that MCQs rely more on educated guesses and require less in-depth reasoning. In contrast, CoT prompts often lead to incorrect explanations and lower performance, as models struggle with reasoning or generate hallucinated answers. This pattern reflects human behavior, where selecting a single answer is generally easier than providing a detailed explanation.

## 2 THE LOGICVISTA DATASET

### 2.1 DATA SOURCES

Most multimodal evaluation benchmarks source images from the open internet, which risks data leakage into MLLM training datasets, potentially giving some models an unfair advantage. To ensure the integrity of LogicVista's evaluations, we prevent data leakage by collecting and annotating our samples from licensed IQ test sources, with permission from the test creators. These tests are not publicly accessible and require either payment or registration, significantly reducing the likelihood that LogicVista's samples have been seen by MLLMs during training. Licensing details and sources are also included in the dataset annotations. Additionally, we conducted Google reverse image

searches on 50 randomly sampled LogicVista data points, finding that our dataset is not available online, whereas many samples from existing benchmarks can be found on the open internet (detailed in Appendix K).

## 2.2 ANNOTATION AND DATA COLLECTION GUIDELINES

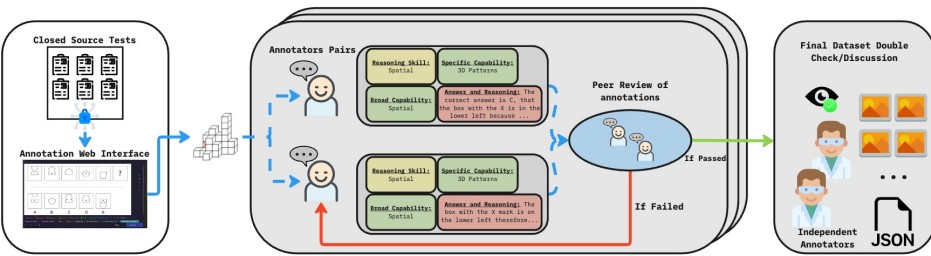

Figure 2: LogicVista's robust manual annotation pipeline ensures high-quality data through multiple rounds of peer review and validation.

**Motivation** LogicVista comprises images designed to assess the underlying reasoning capabilities of MLLMs. Real-life scenes can complicate explicit tests of logical reasoning because they often provide contextual clues that enable an AI agent to deduce answers without engaging in direct reasoning about the scene. To address this, LogicVista features multiple-choice questions across 3 broad capabilities and 11 specific capabilities, clearly specifying the type of reasoning required without the additional context of real-life scenarios. Such question formats are commonly found in intelligence and reasoning tests. Consequently, we initially reviewed over 50 intelligence test distributors for suitable tests and formats, focusing on a diverse range of reasoning categories and test sizes. This process led us to filter down to approximately 10 closed-source test banks, from which we gathered our datasets, seeking permission from the test creators to use their materials for our project.

**Annotation Process** To ensure high-quality annotations, we established a rigorous data collection and annotation pipeline involving six annotators and two project leads, all of whom are STEM students, as detailed in Figure 2. The annotators were organized into pairs, each responsible for annotating the same batch of images. They classified each image based on its logical reasoning, broad capability, and specific capability, while also providing the correct answer and open-ended reasoning annotations. Using an answer key as a reference, annotators developed in-depth explanations for why each answer choice was correct.

To maintain accuracy and consistency in the open-ended reasoning annotations, the teams collaborated to reach a consensus on the correct answers and reasoning for each sample in LogicVista. After each annotation sprint, the teams conducted peer reviews, exchanging and refining their annotations. Suggested edits were merged into a single batch for each group, which was then submitted to project advisors who acted as independent reviewers to ensure the quality of the open-ended reasoning annotations and correct answers. Each batch underwent cross-validation by an independent group of annotators, providing an objective quality check before incorporation into the final LogicVista dataset.

At the end of the project, the group reconvened to verify the robustness of all samples, ensuring that key annotations, such as open-ended reasoning and question classifications, were both accurate and comprehensive. This meticulous process spanned approximately four months. All data were collected and annotated from closed sources requiring payment or registration for access, significantly reducing the likelihood of the dataset being included in prior training or benchmarking datasets, as outlined in Appendix K.

**Annotation Categories** To enable a thorough analysis of MLLM performance on visual logical reasoning tasks, we provided fine-grained data annotations that allow for examination across various aspects. With this goal in mind, we annotated each sample in LogicVista with the following details: the question, the answer, the correct MCQ answer, an open-ended reasoning explanation for why

the MCQ answer is correct, the reasoning skill used, the broad multimodal capability, the specific multimodal capability, and licensing/data source information.

**Annotation Tool** Additionally, we developed an annotation tool, detailed in Appendix L, which we will release for crowdsourcing. This will allow us to scale the pipeline to the broader community, ensuring the sustainability and scalability of LogicVista for future developments.

## 2.3 LOGICVISTA ANALYSIS

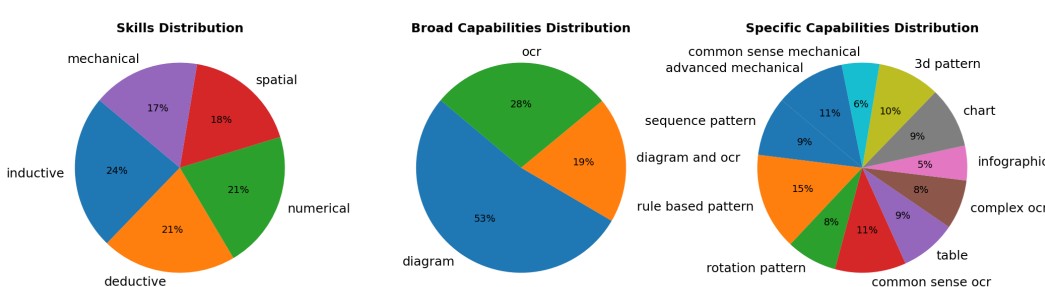

Figure 3: Proportion of reasoning skills and capabilities. On the left is the proportion of questions belonging to each reasoning skill. In the middle is the proportion of questions belonging to broad visual capabilities. On the right is the proportion of questions belonging to specific visual capabilities

A detailed breakdown of the contents in LogicVista is shown in Figure 3. The dataset encompasses 5 core reasoning skills based on fundamental human reasoning capabilities, which we further categorize into broad multimodal capability formats and specific formats for in-depth analysis of MLLM performance in visual logical reasoning. The data is sourced from over 15 human intelligence tests. Samples from the dataset, presented in Appendices I and J, illustrate the richness and diversity of the logical reasoning skills and formats included in LogicVista.

**Multi-modal Capabilities** We define multi-modal capabilities as distinct from reasoning skills, as these capabilities are essential for understanding a multi-modal scene and extracting relevant information. Capabilities refer to the modes in which logical reasoning questions are presented. To ensure comprehensive coverage in LogicVista, we have established a diverse array of *3 broad capabilities* and *11 specific capabilities* for evaluation. This division into broad and specific categories provides hierarchical insights into how well MLLMs perform in areas such as OCR versus diagrams at a broader level, while also offering detailed insights into their performance across specific categories, including various diagram presentation styles and formats. This diversity ensures that LogicVista thoroughly evaluates a wide range of logical situations that an MLLM may encounter in everyday reasoning, providing in-depth insights into each capability. Figure 3 illustrates that LogicVista incorporates a balanced mix of various capabilities, including samples that leverage both to solve problems effectively. We define these capabilities in detail in Appendix B.

**Visual Logical Reasoning Skills** The reasoning skills that were of interest for this benchmark were based on common reasoning skills humans use for critical thinking and problem-solving in most contexts derived from popular human intelligence tests. For our evaluation, we summarize these to include the following 5 skills. As seen in Figure 3, LogicVista contains a wide, balanced range of all core reasoning skills. We define these skills in detail in Appendix B.

## 3 EXPERIMENTS

### 3.1 EVALUATION MODELS

To evaluate the performance of MLLMs on LogicVista, we selected a range of representative models detailed in Appendix C Table. 3. Specifically, we selected 11 models for evaluation, including LLaVA (Liu et al., 2023a; 2024), MiniGPT4 (Zhu et al., 2023), Otter (Li et al., 2023a), variations of OpenAI's GPT-4 (OpenAI et al., 2024), variations of Anthropic's Claude (Anthropic, 2023),

variations of Google's Gemini (Team et al., 2024), BLIP-2 (Li et al., 2023b), and InstructBLIP (Dai et al., 2023) We also specifically included models pix2struct (Lee et al., 2023) as they have been tuned to understand chart or diagram data.

We selected a diverse set of models that represent the current MLLM landscape in both open and closed-source MLLMs. This selection encompasses various model sizes and architectures, incorporating different visual encoders, backbone language models, and training datasets. Our goal was to obtain a comprehensive understanding of MLLMs' performance in visual logical reasoning skills. The breakdown of models we selected for our experiments is detailed in Table 3.

Additionally, we incorporated baseline comparisons to provide a reference for interpreting the results from MLLMs. First, we established a random baseline that selects answer choices by randomly sampling from a Gaussian distribution. We also included a frequentist baseline, which selects the most commonly seen option in the dataset as the response.

## 3.2 EVALUATION PROTOCOLS

We evaluate the models on LogicVista using two setups: **MCQ-based** prompting assessed with an LLM-based answer choice extractor, and **CoT-based** prompting evaluated by an LLM-as-judge. We chose an MCQ-only evaluation strategy for its straightforward and efficient approach to gauging MLLM performance. The binary nature of MCQ answers (correct/incorrect) simplifies grading and allows for easy comparisons across various tasks and datasets. This method is also used by several other datasets, such as MathVista (Lu et al., 2024), establishing its reliability.

However, we recognize that MCQ-only evaluations have limitations, as they obscure the reasoning processes of MLLMs by reducing the evaluation to a binary output without revealing the rationale behind the answers. To address this, we also incorporate a chain-of-thought evaluation format, where we ask an LLM judge (GPT-4o) to analyze CoT responses from MLLMs. This judge compares these responses to the ground truth and explains which aspects were incorrect, providing a finer understanding of whether MLLMs arrive at the correct answer with sound reasoning or if they produce incorrect answers despite valid reasoning.

To calculate accuracy scores for each model, we use different methods depending on whether we are evaluating with the MCQ or CoT approach. For MCQ, an LLM-based extractor isolates the selected answers from the MLLMs' outputs (which are often full sentences rather than single letters) and compares them to the correct answers. In the CoT approach, an LLM judge assesses the open-ended responses against the ground truth. In both cases, the overall logical reasoning score is determined by dividing the number of correct responses by the total number of samples in that particular category, whether it pertains to reasoning skills or capabilities.

## 4 EXPERIMENTAL RESULTS

### 4.1 VISUAL LOGICAL REASONING PERFORMANCE

Table 2 highlights the results for these models across five logical reasoning categories. We analyzed models of different architectures and sizes, comparing them against random and frequentist baselines.

Our results show that most models struggle with inductive, numerical, and spatial reasoning, while generally performing well in deductive and mechanical reasoning tasks.

**Training Limitations**: We believe this disparity arises from the limited exposure visual encoders like CLIP (Dosovitskiy et al., 2021; Radford et al., 2021) have to inductive, numerical, and spatial reasoning scenarios in their training data. These encoders are typically trained on standard computer vision (CV) datasets focused on object recognition, classification, and segmentation using text labels. While this equips models to excel in tasks like identifying and labeling objects or understanding cause-effect relationships, it leaves them ill-prepared for reasoning on spatial dynamics or inductive patterns.

For instance, LLaVA models, often fine-tuned with data capturing object names and coordinates, show stronger spatial, inductive, and deductive reasoning than other open-source counterparts. This underscores the need for vision encoders that capture detailed image information. Despite the

capabilities of advanced backbone LLMs, MLLM reasoning is limited when visual encoders cannot extract crucial logical information. The narrow focus of CV datasets further constrains these models' ability to handle more complex reasoning tasks.

Table 1: LogicVista evaluation results on various multimodal LLMs on broad multi-modal capabilities. Higher scoring models are highlighted **green** and lower scoring models are highlighted **yellow**.

| Model | Broad Capabilities | | | Model | Broad Capabilities | | |
|---|---|---|---|---|---|---|---|
| | Diagram | OCR | Diagram and OCR | | Diagram | OCR | Diagram and OCR |
| Frequentist | 26.69% | 23.20% | 21.84% | pix2struct | 9.60% | 6.60% | 5.75% |
| Random | 22.46% | 24.80% | 22.99% | miniGPTvicuna7B | 11.15% | 9.43% | 6.90% |
| Claude 3.5 Sonnet | 36.02% | 62.40% | 39.08% | miniGPTvicuna13B | 13.00% | 16.98% | 12.64% |
| Claude 3 Opus | 30.51% | 40.80% | 28.74% | instructBLIP-vicuna-7B | 12.07% | 20.28% | 17.24% |
| Claude 3 Sonnet | 30.08% | 48.80% | 29.89% | instructBLIP-vicuna-13B | 10.53% | 13.21% | 14.94% |
| Claude 3 Haiku | 27.12% | 40.80% | 16.09% | instructBLIP-flan-t5-xl | 20.74% | 21.70% | 17.24% |
| GPT4 | 26.63% | 38.68% | 25.29% | instructBLIP-flan-t5-xxl | 20.12% | 25.47% | 18.39% |
| GPT-4o | 33.47% | 47.20% | 26.44% | BLIP2 | 19.50% | 23.11% | 18.39% |
| GPT-4o-mini | 25.85% | 47.20% | 25.29% | LLAVA7B | 29.72% | 27.36% | 26.44% |
| Gemini-Pro | 37.29% | 54.40% | 32.18% | LLAVA13B | 21.67% | 24.06% | 14.94% |
| Gemini-Flash | 34.75% | 45.60% | 24.14% | LLAVANEXT-7B-vicuna | 26.01% | 23.11% | 19.54% |
| otter9B | 23.22% | 22.17% | 18.39% | LLAVANEXT-13B-vicuna | 24.15% | 23.58% | 20.69% |

**Architectural Limitations**: Inductive reasoning often involves identifying patterns across multiple examples, which is not emphasized in standard visual training. In contrast, deductive reasoning—grounded in logical structures and patterns common in textual data—is a strength for LLMs due to their extensive training on large text corpora. Numerical reasoning, another area of weakness, requires an understanding of mathematical principles visually—something multi-modal models struggle to integrate with both visual and textual information. Additionally, the architecture of these models may favor certain reasoning types over others. For instance, while attention mechanisms excel at sequential deduction, they may struggle to effectively capture visual spatial relationships. Ultimately, these challenges in reasoning tasks arise from both the limitations in training data and the architectural design of multimodal LLMs. We further elaborate on these points in Section 4.5.

Table 2: LogicVista evaluation results on various multimodal LLMs on each logical reasoning skill. The higher scoring models are highlighted **green** and lower scoring models are highlighted **yellow**.

| Model | Logical Reasoning Skills | | | | |
|---|---|---|---|---|---|
| | Inductive | Deductive | Numerical | Spatial | Mechanical |
| Frequentist | 25.23% | 19.35% | 27.37% | 26.58% | 25.67% |
| Random | 21.50% | 30.11% | 16.84% | 18.99% | 29.73% |
| Claude 3.5 Sonnet | 27.10% | 65.59% | 47.37% | 29.11% | 52.70% |
| Claude 3 Opus | 21.50% | 49.46% | 26.32% | 25.33% | 45.95% |
| Claude 3 Sonnet | 28.04% | 53.76% | 32.63% | 27.85% | 33.78% |
| Claude 3 Haiku | 24.30% | 47.31% | 15.79% | 24.05% | 33.78% |
| GPT4 | 23.36% | 54.84% | 24.21% | 21.52% | 41.89% |
| GPT-4o | 23.36% | 58.06% | 26.32% | 26.58% | 48.65% |
| GPT-4o-mini | 22.43% | 53.76% | 26.32% | 21.52% | 35.14% |
| Gemini-Pro | 28.97% | 62.37% | 32.63% | 24.05% | 60.81% |
| Gemini-Flash | 32.71% | 51.61% | 25.26% | 20.25% | 50.00% |
| otter9B | 31.78% | 24.73% | 18.95% | 18.99% | 21.62% |
| pix2struct | 12.15% | 6.45% | 2.11% | 7.59% | 17.57% |
| miniGPTvicuna7B | 10.28% | 9.68% | 7.37% | 3.80% | 27.03% |
| miniGPTvicuna13B | 13.08% | 23.66% | 10.53% | 10.13% | 17.57% |
| instructBLIP-vicuna-7B | 4.67% | 21.51% | 24.21% | 2.53% | 22.97% |
| instructBLIP-vicuna-13B | 3.74% | 10.75% | 18.95% | 5.06% | 17.57% |
| instructBLIP-flan-t5-xl | 23.36% | 22.58% | 22.11% | 7.59% | 33.78% |
| instructBLIP-flan-t5-xxl | 17.76% | 30.11% | 24.21% | 20.25% | 22.97% |
| BLIP2 | 17.76% | 23.66% | 23.16% | 24.05% | 18.92% |
| LLAVA7B | 29.91% | 29.03% | 26.32% | 25.32% | 36.49% |
| LLAVA13B | 18.69% | 31.18% | 20.00% | 27.85% | 24.32% |
| LLAVANEXT-7B-vicuna | 26.17% | 21.51% | 25.26% | 27.85% | 29.73% |
| LLAVANEXT-13B-vicuna | 22.43% | 22.58% | 26.32% | 26.58% | 25.68% |
| LLAVANEXT-7B-mistral | 16.82% | 34.41% | 23.16% | 21.52% | 22.97% |
| LLAVANEXT-34B-NH | 20.56% | 52.69% | 30.53% | 24.05% | 40.54% |

## 4.2 VISUAL CAPABILITIES PERFORMANCE

We highlight the performance of MLLMs on various broad and specific visual capabilities in Appendix J, Tables 1, 4, and 5.

**Broad Capabilities** Our results show that, on average, most models perform better on OCR-type questions than on diagram-format questions within the broad capability category.

A possible reason why multi-modal LLMs (MLLMs) perform better on OCR-type questions compared to diagram-based questions is the difference in visual reasoning complexity. OCR tasks mainly involve recognizing and extracting textual information from images, which plays to the strengths of visual encoders in object recognition and classification. The text in OCR tasks is typically structured, with clear boundaries and minimal need for spatial or abstract reasoning. This allows the MLLM to focus on straightforward text recognition, followed by reasoning using the LLM backbone, which has been shown to excel at various textual reasoning tasks (liu et al., 2023; OpenAI et al., 2024; Touvron et al., 2023). As a result, the multi-modal reasoning task is simplified into a more manageable text-based reasoning process.

In contrast, diagram-based questions typically demand more complex spatial reasoning, pattern recognition, and an understanding of relationships between visual elements. These tasks go beyond merely recognizing objects or labels, requiring the ability to interpret how objects interact, and their relative positions, and sometimes even apply inductive or deductive reasoning. Visual encoders, often not optimized for spatial or abstract relationships, tend to struggle with these challenges. The complexity of interpreting geometric shapes, spatial arrangements, and abstract concepts in diagrams is much greater than the more straightforward task of recognizing and interpreting text in OCR scenarios, as it requires more than simple recognition and identification of basic relationships.

**Specific Capabilities** We found that MLLMs generally perform well on tasks involving complex OCR, common sense OCR, advanced mechanical reasoning, common sense mechanical reasoning, and rotational patterns. However, they tend to struggle with tasks that require understanding 3D patterns and sequential patterns.

This reinforces our earlier hypothesis that MLLMs excel in OCR and mechanical reasoning tasks because these visual formats primarily focus on recognizing simple relationships and identifying objects, rather than interpreting complex spatial interactions. Mechanical formats often depict real-life scenes, making it easier to discern relationships compared to abstract patterns, where the spatial relationships in 3D and sequential formats are more challenging to extract.

In contrast, tasks involving 3D and sequential pattern recognition require a more nuanced understanding of spatial relationships, movement, and order—capabilities that may be underdeveloped in these models due to limitations in their training data and architectures. Spatial and sequential diagram-based tasks, as well as 3D reasoning, demand an advanced ability to comprehend spatial hierarchies and continuous pattern changes—areas where current visual encoders typically struggle. This lack of spatial depth and temporal awareness contributes to the weaknesses observed in MLLMs when addressing more complex reasoning scenarios.

## 4.3 CASE STUDIES ON LOGICVISTA

Our case studies (Figure 4, 9, 10) show that these errors often occur because MLLMs overlook important details or hallucinate facts, yet still guess the correct solution. This underscores the need for better visual encoders that can capture intricate spatial details beyond recognition. In our MiniGPT-4 case study (Figure 4), while the model reaches the correct answer, the left-hand example reveals a failure to grasp key spatial relationships, guessing "C" simply because the question mark is unfilled. This likely stems from the visual encoder's limitations in interpreting intricate spatial details. Conversely, in the right-hand example, hallucinations lead to incorrect reasoning. Similarly, MiniGPT-4 fabricates details about pipe sizes, resulting in inaccurate reasoning despite correctly identifying certain image elements. Closed-source flagship models also suffer from these visual encoder limitations, as seen in our SOTA model case studies in Appendix F. We also conduct a more in-depth case study analysis of vision encoder performance of MLLMs in Appendix F.

## 4.4 FINE-GRAINED ANSWER ANALYSIS

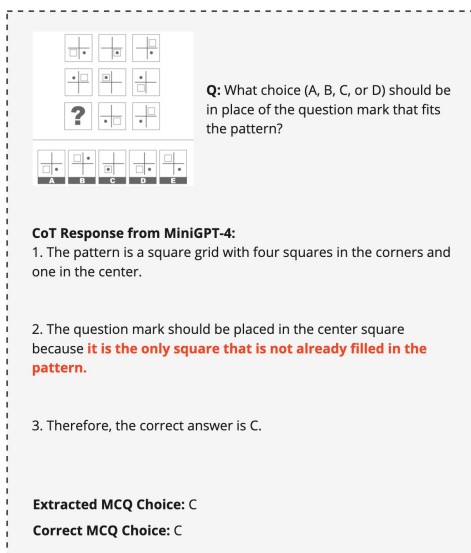
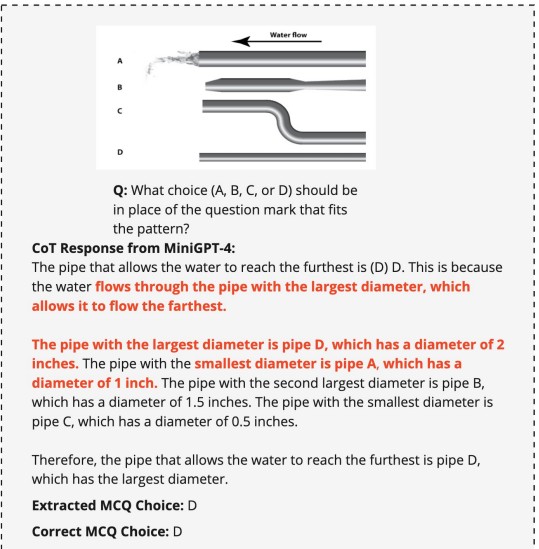

Figure 4: Case study of MiniGPT-4 outputs shows both examples providing correct answers but with flawed CoT reasoning. On the left, the model fails to capture the spatial pattern—where the dot stays constant and the square shifts right—leading to answer C. On the right, the model hallucinates facts about the pipes, missing the key detail that narrower pipes result in faster exit velocity, making D the correct choice.

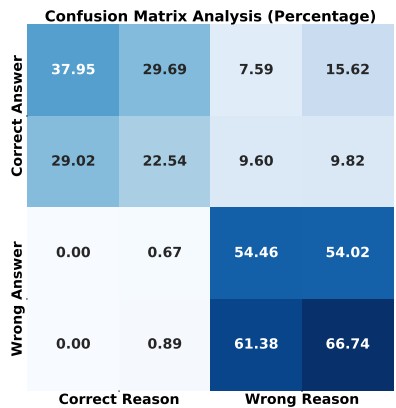

Figure 5: Confusion matrix for performances of SOTA flagship models: Claude 3.5 Sonnet, Gemini Pro, GPT-4o, GPT-4o-mini, arranged clockwise from the top left in each cell.

Using chain-of-thought (CoT) evaluations, where an LLM acts as the judge to compare MLLM outputs to ground truth reasoning, we find that most incorrect responses from MLLMs arise from both incorrect answers and flawed reasoning. This suggests the models either fail to fully understand the problem or miss critical details needed to answer accurately. Interestingly, many correct answers still exhibit faulty reasoning, as shown in Figure 5. For example, Claude 3.5 Sonnet, the top-performing model across reasoning skills and capabilities, gave incorrect answers with faulty explanations 54% of the time and correctly answered questions with incorrect explanations 7.9% of the time. Other models, such as Gemini-Pro and GPT-4, displayed similar patterns, with Gemini Pro leading in faulty explanations at 16%. A manual review of Claude 3.5 Sonnet's responses revealed that over 46% of its outputs included some form of hallucination, either about the image contents (e.g., fabricating facts about patterns or sequences) or general knowledge (e.g., physics, deductive reasoning). Overall, GPT-4o-mini performs the worst, frequently providing incorrect answers and explanations, while Claude 3.5 Sonnet achieves the best results. We observed several instances where models gave correct answers but incorrect explanations. We perform an additional analysis of how MCQ-based evaluation affects evaluation accuracy compared to CoT-based evaluations in Appendix G.

## 4.5 MODEL COMPARISONS

**Vision Component**: In our evaluation, we considered only open-source vision models for benchmarking. The primary vision encoders used were CLIP-ViT (428M) and EVA-ViT-G (1.13B). When paired

with large language models (LLMs) such as Vicuna 7B and 13B, the LLaVA variants incorporating CLIP-ViT demonstrated superior performance in spatial, deductive, and inductive reasoning tasks compared to InstructBLIP, which utilized the EVA-ViT-G encoder. Despite these observations, it is challenging to declare a definitive superior model for logical reasoning due to the absence of a comprehensive ablation study, which would provide a more thorough analysis of the strengths and weaknesses of each model configuration. We also present a case study of the failures of modern visual encoders in Appendix F.

**Language Modeling Component**: A range of LLMs, including Vicuna, Flan-T5, and LLaMA, were tested by LogicVista to evaluate their performance. With EVA-ViT-G as the vision encoder, the BLIP-2-12B model combined with Flan-T5-XXL outperformed InstructBLIP, which used Vicuna-13B, particularly in spatial reasoning tasks. This suggests that the Flan-T5 model may possess stronger spatial language processing capabilities compared to Vicuna-13B. This observation highlights the potential impact of different LLMs on the effectiveness of multimodal systems. The performance difference indicates that the choice of LLM can significantly affect the effectiveness of multimodal systems in specific reasoning tasks. Flan-T5's demonstrated strength in spatial reasoning underscores the importance of choosing LLMs that align with specific reasoning capabilities. However, a more detailed analysis of how LLM logical reasoning performance relates to multimodal logical reasoning could provide better insights into how different LLMs impact the overall performance of visual reasoning systems.

**Training Data**: The comparison of training data performance reveals that MiniGPT, with its datasets including CC3m, SBU, LAION-400M, and a custom set of 3500 images, excels particularly in induction tasks. This suggests that MiniGPT's training data could be highly effective for tasks requiring the model to generalize from specific inductive examples to broader patterns. On the other hand, InstructBLIP's training data, which encompasses BLIP2 and 26 transformed datasets, shows stronger performance in a broader range of evaluation categories. This indicates that the suitability of training data may vary depending on the specific types of reasoning or tasks. Some of these datasets may have more samples covering specific reasoning tasks causing different datasets to provide distinct advantages for different reasoning skills.

**Closed/Open-Source Models**: The results suggest that closed-source models like GPT, Gemini, and Claude significantly outperform open-source models in deduction and mechanical reasoning, often with double the accuracy. This advantage likely stems from proprietary optimizations, training techniques, model size, or undisclosed data. Additionally, the continuous updates and fine-tuning specific to these models may contribute to their superior performance. However, in numerical, spatial, and inductive reasoning tasks, both open- and closed-source models show similar effectiveness, with accuracy rates between 22% and 31% across leading closed-source models (GPT, Claude, Gemini) and open-source models (13B LLaVA, Yi models, InstructBLIP). While closed-source models excel in deduction and mechanical reasoning, both model types struggle similarly with spatial and inductive reasoning, suggesting the challenges lie more in the fundamental limitations of current MLLM technologies for visual logical reasoning than in proprietary enhancements. Greater transparency and research could clarify these performance differences and inform future advancements in both open-source and closed-source models, potentially bridging the gap in reasoning capabilities.

## 5 CONCLUSION

In this work, we introduce LogicVista, a comprehensive benchmark designed to evaluate MLLM performance on complex visual logical reasoning tasks, covering inductive, deductive, spatial, numerical, and mechanical visual reasoning skills. We assess 11 state-of-the-art open and closed-source MLLMs, offering insights into the current landscape of these models. Our detailed analysis reveals that MLLMs often struggle with intricate spatial and logical details in images, as their visual encoders are typically trained for broad object recognition. This focus leads to failures in tasks that require a deep understanding of spatial relationships, particularly in inductive, spatial, and numerical reasoning. Our fine-grained CoT case study underscores this limitation, showing that MLLMs tend to generalize rather than capture precise spatial details in both abstract and real-life scenes. We also find that MCQ evaluations often overestimate MLLM performance, as they fail to assess reasoning as effectively as CoT methods. Therefore, we propose future benchmarks emphasize open-ended evaluations that assess the reasoning process, not just final answers.

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

# Appendix: LogicVista: Multimodal LLM Logical Reasoning Benchmark in Visual Contexts

## LIMITATIONS

A limitation of our work is the absence of a human baseline for comparison. Having this baseline would offer valuable insights into how MLLMs perform relative to humans. Although several of our test sources suggest that average human performance is around 75%, this figure varies across different reasoning skills, which is why we chose not to include it in our study for the sake of reliability and accuracy. A more dependable approach would be to conduct multiple human trials to establish a consistent average performance for comparison.

Additionally, while our dataset size is comparable to other multimodal benchmarks like MM-vet (Yu et al., 2023), it is relatively smaller than some larger-scale benchmarks such as MMBench or MMMU (Liu et al., 2023c; Yue et al., 2024). To address this, we will release a crowdsourcing annotation tool, detailed in Appendix L, to further scale LogicVista in the future.

To address both concerns and promote further research, we have also open-sourced these reasoning annotations. They are now publicly available for the community, providing a valuable resource for training and improving the logical reasoning capabilities of multimodal LLMs. We encourage future work to make full use of these annotations to develop more comprehensive and contextually rich evaluation methods.

## A  RELATED WORKS

**LLM-Based Evaluation.** LogicVista adopts an open-ended LLM-based evaluation approach, which facilitates the generation and assessment of diverse answer styles and question types beyond the limitations of binary or multiple-choice responses. This innovative method leverages the capabilities of large language models (LLMs) for comprehensive model evaluation, a technique that has been effectively applied in natural language processing (NLP) tasks and other VQA benchmarks (Chiang & yi Lee, 2023; Liu et al., 2023b; Fu et al., 2023b; Jin et al., 2024; Lu et al., 2024). Our findings show that this LLM-based evaluation framework is both versatile and robust, providing a unified and flexible assessment across different modalities, including open- and closed-ended responses. By accommodating a broad range of answer styles and question types, this approach deepens and expands model evaluation, leading to a more comprehensive understanding of model performance.

**Vision-Language Benchmarks** Traditional vision-language benchmarks have largely focused on evaluating specific perceptual abilities. Datasets like MM-vet, RAVEN, CLEVR-X, and TextVQA each address distinct aspects of visual recognition: TextVQA emphasizes recognition-based VQA, testing how well models can caption and accurately describe key image details; MM-vet evaluates world knowledge, basic math, detail capture, and OCR in recognition tasks and everyday scene reasoning. Meanwhile, RAVEN and CLEVR-X assess spatial relation recognition in 2D and 3D objects, providing insights into how well MLLMs understand spatial reasoning (Goyal et al., 2017b; Yu et al., 2023; Zhang et al., 2019; Singh et al., 2019a; Sidorov et al., 2020; Salewski et al., 2022). Image captioning and description generation have also been extensively studied (Chen et al., 2015; Agrawal et al., 2019), along with more specialized tasks like scene text understanding (Singh et al., 2019b; Sidorov et al., 2020; Yang et al., 2020) and integrating external knowledge (Marino et al., 2019). Other benchmarks, such as OlympiadBench (He et al., 2024), focus on Olympiad-level math and science challenges to compare MLLMs with human performance. Large-scale multidisciplinary benchmarks like MMMU (Yue et al., 2024) assess MLLMs across a range of subjects, including science, math, humanities, and history.

## B  DATASET DEFINITIONS

Here we define concretely what each of our capabilities and logical reasoning skill categories refer to.

## B.1 LOGICAL REASONING SKILLS

We define our 5 **logical reasoning skills** based on common human visual reasoning abilities. Our goal is to assess how effectively MLLMs perform in general reasoning skills that humans rely on for everyday problem-solving. These skills reflect the types of reasoning that MLLM agents are likely to encounter in real-world settings where they may be deployed. Our definitions are largely inspired by traditional human IQ and intelligence tests.

- **Inductive Reasoning** the ability to infer the next entry in a pattern given a pattern of observations. It is the ability to make generalizations based on some observations and make an educated guess. It moves from many specific observations to a generalization. An example could be given observations that when John eats dairy products, he gets a stomach ache. An inductive conclusion can be drawn that he is most likely lactose intolerant.

- **Deductive Reasoning** the ability to conclude a specific case when given a general principle or pattern. It moves from the general to the specific. An example could be given the statement "all men are mortal", one can conclude that "John is mortal" because John is a man.

- **Numerical Reasoning** the ability to read arithmetic problems in the image and solve the math equations. An example could be given the equation "10 + 10 = ?", the answer would be "20".

- **Spatial Reasoning** the ability to understand the spatial relationship between objects and patterns and reason with those relationships. An example could be seeing an unfolded box and understanding what the box could look like when it is folded up.

- **Mechanical Reasoning** the ability to recognize a physical system and solve equations based on that system or answer questions about that system. An example could be seeing a set of 3 gears and understanding which gears will turn clockwise and which ones will turn counterclockwise.

## B.2 BROAD AND SPECIFIC CAPABILITIES

We categorize our multi-modal capabilities into broad and specific classifications to gain hierarchical insights into which information formats are better or worse understood by MLLMs.

Here we present our definitions for **broad capabilities**:

- **Optical Character Recognition (OCR):** refers to the ability to reason over text inside images and scenes.

- **Diagrams:** refers to the ability to reason about diagrams that represent real-life scenes, abstract logic, spatial relationships, and more.

- **Mixed (Both OCR and Diagram):** refers to an integration of both OCR and diagrams, where comprehending the text and the visual elements within the image is essential for accurately answering the question.

Here we present our definitions for **specific capabilities**:

- **Chart:** refers to numerical charts and graphs.

- **Infographic:** refers to infographic-style puzzles that illustrate both real-life and abstract scenes.

- **Table:** refers to words and numbers only tables depicting some trend or concept.

- **Common Sense OCR:** refers to text questions describing common everyday situations using common English words.

- **Complex OCR:** refers to text questions describing technical or highly abstract situations using jargon and complex sentences.

- **Rotation Pattern:** Patterns and puzzles that necessitate an understanding of 2D and/or 3D object rotations.

- **3D Pattern:** Patterns that require 3D spatial relation understandings.

- **Rule Based Pattern:** Patterns that require understanding of a set of externally defined rules.

- **Sequence Pattern:** Patterns presented in a strictly sequential format, typically involved with induction.

- **Common Sense Mechanical:** Puzzles concerned with a common sense understanding of basic physics and mechanics.

- **Advanced Mechanical:** Puzzles concerned with an advanced and specialized understanding of physics and mechanics.

## C  SELECTED MLLMs FOR EVALUATION

| Model | Size | Language Model | Vision Model |
|---|---|---|---|
| Claude 3.5 Sonnet | N/A[1] | N/A | N/A |
| Claude 3 Opus | N/A | N/A | N/A |
| Claude 3 Sonnet | N/A | N/A | N/A |
| Claude 3 Haiku | N/A | N/A | N/A |
| GPT-4 Vision | N/A | N/A | N/A |
| GPT-4o | N/A | N/A | N/A |
| GPT-4o-mini | N/A | N/A | N/A |
| Gemini Pro | N/A | N/A | N/A |
| Gemini Flash | N/A | N/A | N/A |
| Otter-9B | 9B | MPT-7B | CLIP ViT-L/14 |
| Pix2Struct | 1.3B | ViT | ViT |
| MiniGPT-4-7B | 7B | Vicuna-7B | BLIP-2 Q-Former |
| MiniGPT-4-13B | 13B | Vicuna-13B | BLIP-2 Q-Former |
| InstructBLIP-Vicuna-7B | 7B | Vicuna-7B | BLIP-2 Q-Former |
| InstructBLIP-Vicuna-13B | 13B | Vicuna-13B | BLIP-2 Q-Former |
| InstructBLIP-FLAN-T5-xl | 3B | FLAN-T5 XL | BLIP-2 Q-Former |
| InstructBLIP-FLAN-T5-xxl | 11B | FLAN-T5 XXL | BLIP-2 Q-Former |
| BLIP-2 | 2.7B | OPT-2.7B | EVA-ViT-G |
| LLaVA-Vicuna-7B | 7B | Vicuna-7B | CLIP ViT-L/14 |
| LLaVA-Vicuna-13B | 13B | Vicuna-13B | CLIP ViT-L/336px |
| LLaVA-NeXT-Mistral-7B | 7B | Mistral-7B | CLIP ViT-L/14 |
| LLaVA-NeXT-Vicuna-7B | 7B | Vicuna-7B | CLIP ViT-L/14 |
| LLaVA-NeXT-Vicuna-13B | 13B | Vicuna-13B | CLIP ViT-L/336px |
| LLaVA-NeXT-Nous-Hermes-Yi-34B | 34B | Nous Hermes 2-Yi-34B | CLIP ViT-L/336px |

Table 3: Summary of the MLLMs used for evaluations in this study. Model details for close-sourced models like Claude, GPT, and Gemini are not open to the public.

## D  BROAD AND SPECIFIC VISUAL CAPABILITIES EVALUATION

We present tabular results evaluating various SOTA open-source and closed-source MLLM models in Table 1, 4, and 5, analyzing their performance across different visual capabilities.

## E  SOTA MODEL EVALUATION RESULT

We present graphs illustrating the evaluations of key SOTA closed-source flagship models. Our analysis shows that Claude 3.5 Sonnet consistently performs well across all categories of reasoning and capabilities, with GPT-4o and Gemini Pro following closely in second place.

---

[1]N/A: Not disclosed

Table 4: Model evaluation results on various multimodal LLMs for Specific Capabilities (Part 1). The highest scoring models are highlighted **green** and lower scoring models are highlighted **yellow**.

| Model | 3D Pattern | Rule Based Pattern | Sequence Pattern | Rotation Pattern | Table | Chart |
|---|---|---|---|---|---|---|
| Frequentist | 27.91% | 26.87% | 17.07% | 31.43% | 30.77% | 23.81% |
| Random | 23.26% | 20.90% | 21.95% | 14.29% | 7.69% | 23.81% |
| Claude 3.5 Sonnet | 25.58% | 32.84% | 17.07% | 34.29% | 51.28% | 50.00% |
| Claude 3 Opus | 16.28% | 25.37% | 21.95% | 28.57% | 17.95% | 33.33% |
| Claude 3 Sonnet | 27.91% | 31.34% | 21.95% | 28.57% | 35.90% | 40.48% |
| Claude 3 Haiku | 23.26% | 22.39% | 24.39% | 28.57% | 25.64% | 14.29% |
| GPT4 | 25.58% | 26.87% | 12.20% | 20.00% | 28.21% | 26.19% |
| GPT-4o | 27.91% | 28.36% | 19.51% | 20.00% | 17.95% | 30.95% |
| GPT-4o-mini | 27.91% | 23.88% | 19.51% | 14.29% | 23.08% | 33.33% |
| Gemini-Pro | 23.26% | 26.87% | 31.71% | 25.71% | 33.33% | 38.10% |
| Gemini-Flash | 13.95% | 38.81% | 21.95% | 28.57% | 25.64% | 33.33% |
| otter9B | 11.63% | 37.31% | 24.39% | 25.71% | 28.21% | 11.90% |
| pix2struct | 4.65% | 7.46% | 17.07% | 14.29% | 7.69% | 0.00% |
| miniGPTvicuna7B | 4.65% | 8.96% | 12.20% | 2.86% | 10.26% | 7.14% |
| miniGPTvicuna13B | 11.63% | 14.93% | 12.20% | 5.71% | 10.26% | 14.29% |
| instructBLIP-vicuna-7B | 4.65% | 5.97% | 2.44% | 0.00% | 23.08% | 23.81% |
| instructBLIP-vicuna-13B | 4.65% | 4.48% | 7.32% | 0.00% | 15.38% | 21.43% |
| instructBLIP-flan-t5-xl | 9.30% | 26.87% | 4.88% | 20.00% | 28.21% | 19.05% |
| instructBLIP-flan-t5-xxl | 23.26% | 19.40% | 14.63% | 17.14% | 33.33% | 21.43% |
| BLIP2 | 20.93% | 17.91% | 14.63% | 31.43% | 28.21% | 16.67% |
| LLAVA7B | 27.91% | 32.84% | 24.39% | 22.86% | 23.08% | 21.43% |
| LLAVA13B | 27.91% | 16.42% | 24.39% | 25.71% | 25.64% | 19.05% |
| LLAVANEXT-7B-vicuna | 34.88% | 26.87% | 19.51% | 25.71% | 30.77% | 19.05% |
| LLAVANEXT-13B-vicuna | 27.91% | 22.39% | 17.07% | 31.43% | 30.77% | 21.43% |
| LLAVANEXT-7B-mistral | 13.95% | 14.93% | 21.95% | 28.57% | 20.51% | 23.81% |
| LLAVANEXT-34B-NH | 27.91% | 19.40% | 24.39% | 17.14% | 28.21% | 19.05% |

Table 5: Model evaluation results on various multimodal LLMs for Specific Capabilities (Part 2). The highest scoring models are highlighted **green** and lower scoring models are highlighted **yellow**.

| Model | Infographic | Complex OCR | Common Sense OCR | Advanced Mechanical | Common Sense Mechanical |
|---|---|---|---|---|---|
| Frequentist | 20.83% | 17.65% | 22.45% | 25.00% | 26.92% |
| Random | 16.67% | 35.29% | 30.61% | 25.00% | 38.46% |
| Claude 3.5 Sonnet | 37.50% | 73.53% | 63.27% | 50% | 57.69% |
| Claude 3 Opus | 29.17% | 58.82% | 46.94% | 37.50% | 64.54% |
| Claude 3 Sonnet | 16.67% | 55.88% | 55.10% | 25.00% | 50.00% |
| Claude 3 Haiku | 12.50% | 47.06% | 48.98% | 25.00% | 50.00% |
| GPT4 | 67.65% | 53.06% | 31.25% | 61.54% | 31.25% |
| GPT-4o | 73.08% | 35.42% | 73.08% | 73.08% | 73.08% |
| GPT-4o-mini | 52.94% | 63.27% | 29.17% | 46.15% | 46.15% |
| Gemini-Pro | 58.82% | 69.39% | 58.33% | 65.38% | 65.38% |
| Gemini-Flash | 8.33% | 50.00% | 59.18% | 41.67% | 65.38% |
| otter9B | 29.41% | 20.41% | 22.92% | 19.23% | 19.23% |
| pix2struct | 8.16% | 16.67% | 19.23% | 19.23% | 19.23% |
| miniGPTvicuna7B | 31.25% | 19.23% | 19.23% | 19.23% | 19.23% |
| miniGPTvicuna13B | 32.35% | 20.41% | 16.67% | 19.23% | 19.23% |
| instructBLIP-vicuna-7B | 32.35% | 16.33% | 25.00% | 19.23% | 19.23% |
| instructBLIP-vicuna-13B | 17.65% | 6.12% | 25.00% | 19.23% | 19.23% |
| instructBLIP-flan-t5-xl | 32.35% | 16.33% | 25.00% | 19.23% | 50.00% |
| instructBLIP-flan-t5-xxl | 29.41% | 28.57% | 25.00% | 19.23% | 19.23% |
| BLIP2 | 17.65% | 30.61% | 22.92% | 19.23% | 11.54% |
| LLAVA7B | 31.25% | 26.53% | 31.25% | 46.15% | 46.15% |
| LLAVA13B | 18.75% | 34.62% | 18.75% | 34.62% | 34.62% |
| LLAVANEXT-7B-vicuna | 23.53% | 22.45% | 27.08% | 34.62% | 34.62% |
| LLAVANEXT-13B-vicuna | 20.83% | 20.41% | 20.83% | 34.62% | 34.62% |
| LLAVANEXT-7B-mistral | 31.25% | 18.75% | 30.77% | 34.62% | 30.77% |
| LLAVANEXT-34B-NH | 55.88% | 59.18% | 41.67% | 38.46% | 38.46% |

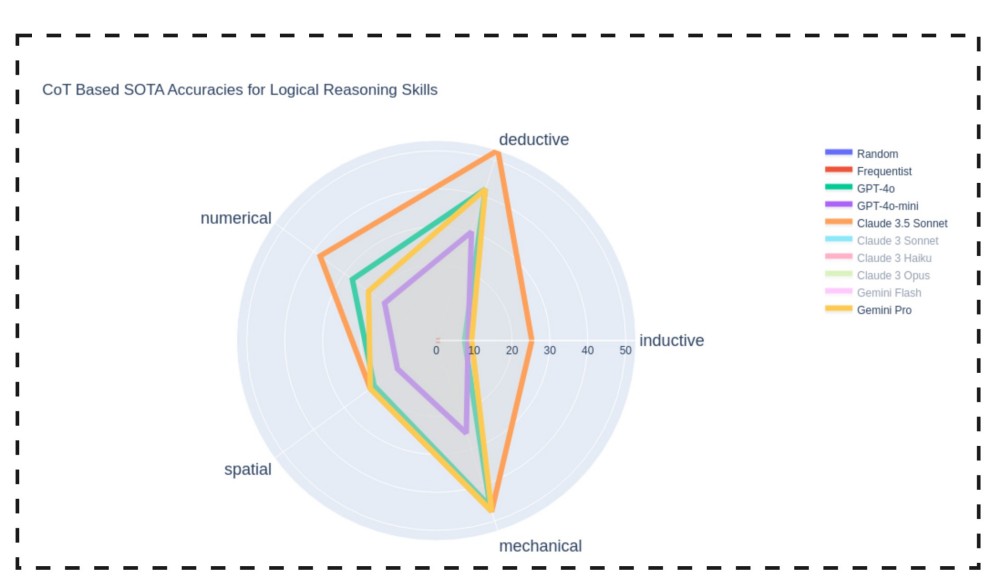

Figure 6: SOTA evaluation results of CoT evaluations on logical reasoning skills. As seen here, Claude 3.5 Sonnet has superior performance.

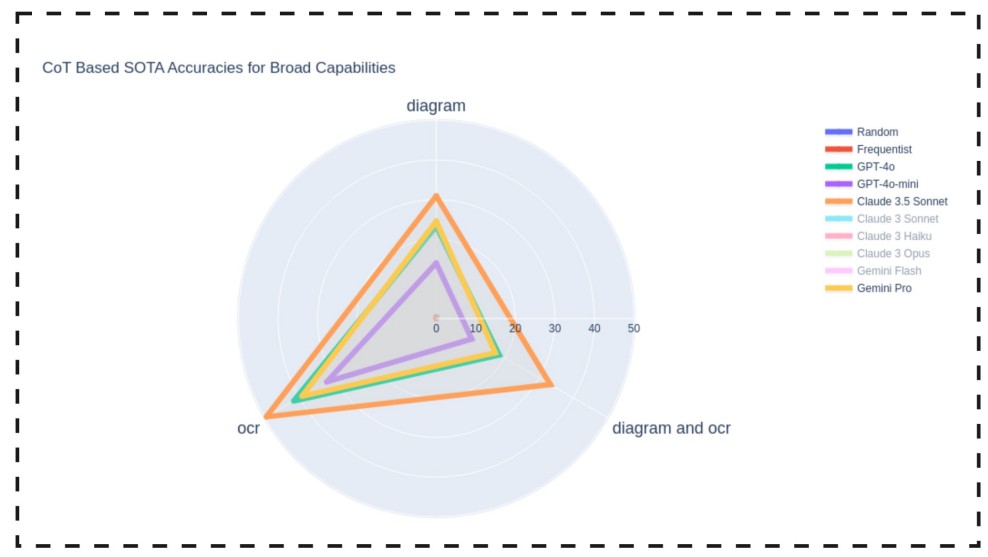

Figure 7: SOTA evaluation results of CoT evaluations on broad visual capabilities. As seen here, Claude 3.5 Sonnet has superior performance.

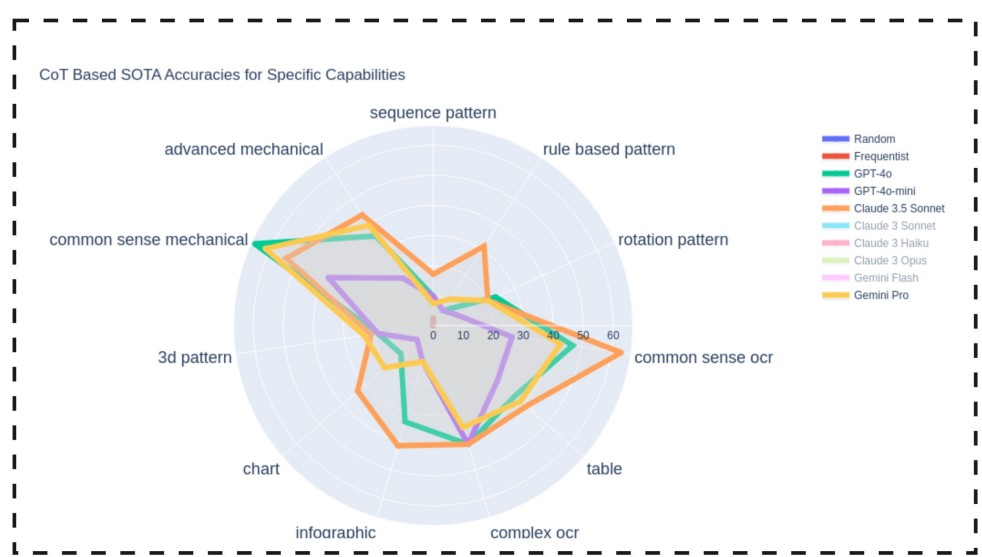

Figure 8: SOTA evaluation results of CoT evaluations on specific visual capabilities. As seen here, Claude 3.5 Sonnet has superior performance. However, it as bested by GPT-4o in some categories like common sense mechanical formats and complex OCR.

## F  SOTA CLOSED-SOURCE MLLMS CASE STUDIES

Given that Claude 3.5 Sonnet is the top-performing MLLM across multiple of LogicVista's question types and formats, we conducted a case study to examine its shortcomings in capturing the spatial and logical relationships essential for complex visual reasoning.

In Figures 9 and 10, Claude 3.5 Sonnet struggles to identify key spatial relationships, such as the shape and number of stars, while also overlooking the logical context necessary for answering LogicVista questions in a CoT format. Similarly, in Figure 10, Claude misinterprets the sequential movement between the circle and triangle. These examples highlight a common issue with modern MLLM vision encoders: they tend to focus on object recognition rather than understanding the relationships between objects, which is essential for accurate visual logical reasoning.

Building on this insight, we conducted an additional experiment to assess the specific details a modern SOTA MLLM can capture. Using Claude 3.5 Sonnet, the top-performing model for LogicVista, we prompted it to provide detailed descriptions of LogicVista samples with a focus on spatial relationships between objects. As shown in Figure 11, MLLMs excel at simpler recognition tasks, such as identifying spring lengths, enabling the model to solve the problem easily. However, when tasked with recognizing more complex spatial relationships, current MLLMs struggle. For instance, in Figure 12, Claude misses intricate spatial patterns and instead focuses on broad features—reflecting a limitation of traditional CV encoders, which are good at general visual recognition but struggle to accurately capture specific spatial arrangements like the positioning of hexagons, squares, and circles. Even in less abstract cases, such as the one on the right in Figure 12 depicting a sled, Claude fails to distinguish key details like the width and size of runners, instead hallucinating differences in sled sizes. When asked specifically about the runner sizes, Claude either misidentifies them as similar or fabricates relationships. This demonstrates the need for vision encoders to be able to capture more intricate spatial details and focus less on recognition, which it already excels at but focus rather more on extracting these key spatial and visual-logical relations.

**CoT Response from Claude Sonnet 3.5:**
To determine which set the Figure belongs to, we need to analyze the characteristics of Set A, Set B, and the Figure:

1. **Set A:**
    - **Contains 5 boxes**
    - **Each box has a mix of different sized stars**
    - **The number of stars in each box varies, but generally includes 4-7 stars**
2. **Set B:**
    - **Contains 5 boxes**
    - **Most boxes have a mix of different sized stars**
    - **The first box contains only one large star**
    - **The number of stars in each box varies, generally 3-6 stars**
3. **The Figure:**
    - **Contains exactly 4 stars**
    - **All stars are of similar size**
    - **Stars are arranged in a 2x2 grid pattern**

Comparing the Figure to both sets:
- **It doesn't match the typical pattern of Set A, which usually has more stars and varying sizes.**
- It doesn't fit with Set B either, as it lacks the variation in star sizes seen in most boxes of Set B.
- The Figure's uniform arrangement and consistent star size is not characteristic of either set.

Therefore, the correct answer is:
(C) Neither set A nor set B
The Figure has its own unique characteristics that don't align with the patterns observed in either Set A or Set B.

**Extracted MCQ Choice:** C

**Correct MCQ Choice:** A

Figure 9: An example of Claude 3.5 Sonnet's failure to capture intricate spatial relationships is shown here. The model recognizes the differently shaped stars but misses the key relationships needed to solve the problem. It also focuses on irrelevant details, overlooking the fact that A contains an even number of stars and B an odd number, which is essential for determining the correct solution.

**Q:** Which set does the Figure belong to? Select from A, B, and C. (A) Set A (B) Set B (C) Neither set A nor set B

**CoT Response from Claude Sonnet 3.5:**

To determine which box comes next in the sequence, let's analyze the pattern in the given sequence:

1. The sequence alternates between triangles pointing up (▲) and triangles pointing right (▶).
2. The circle(s) move position in each step:
    - Start in top-left corner
    - Move to top-right corner
    - **Split into two circles in top-right and top-left corners**
    - **Remain as two circles but move to top-right corner**
    - **Return to a single circle in top-right corner**
3. The last box in the given sequence has a triangle pointing up (▲) and **a single circle in the top-right corner.**

**Following this pattern, the next box should have:**
  - **A triangle pointing right (▶)**
  - **Two circles in the top-right corner**

Looking at the options A through E:

A: Has the correct triangle, but no circles B: Has the wrong triangle and wrong circle position C: Has the correct triangle, but only one circle in the wrong position D: Has the wrong triangle and wrong number of circles E: Has the correct triangle (▶) and the correct number and position of circles

Therefore, the correct answer is E. It continues the established pattern by having a right-pointing triangle and two circles in the top-right corner.

**Extracted MCQ Choice:** E

**Correct MCQ Choice:** C

Figure 10: Another example of Claude 3.5 Sonnet's failure to capture detailed visual logic is its hallucination of spatial facts about the relationship between the triangle and the circle. Claude incorrectly assumes that the circle is splitting in half, when in reality, this appearance results from a sequential relationship between the images.

**CoT Response from Claude Sonnet 3.5:**

This image shows four different coil springs labeled A through D, arranged side by side from left to right. Let me describe each in detail:

**Spring A:**
- **Approximately 10-11 coils**
- **Medium height**
- **Medium coil diameter**
- **Uniform spacing between coils**

**Spring B:**
- **Approximately 8-9 coils**
- **Shortest of the four springs**
- **Similar coil diameter to A**
- **Uniform spacing between coils**

**Spring C:**
- **Approximately 14-15 coils**
- **Tallest spring**
- **Similar coil diameter to A and B**
- **Uniform spacing between coils**
- **Appears slightly thinner in wire gauge**

**Spring D:**
- **Approximately 12-13 coils**
- **Second tallest spring**
- **Similar coil diameter to others**
- **Uniform spacing between coils**
- **Some slight irregularity at the top coil**

Spatial relationships:
- Springs are arranged in ascending order of height from B to C
- Spring A breaks this pattern, being taller than B but shorter than C
- All springs have similar diameters
- The springs are evenly spaced horizontally
- Each spring is labeled with a letter beneath it

All springs appear to be drawn in a somewhat sketchy, hand-drawn style with some minor irregularities in the lines, giving them a more realistic rather than perfectly geometric appearance.

The overall arrangement suggests this might be a comparison or educational diagram showing different spring configurations or sizes.

Figure 11: Another example of Claude 3.5 Sonnet's failure to capture detailed visual logic is its hallucination of spatial facts about the relationship between the triangle and the circle. Claude incorrectly assumes that the circle is splitting in half, when in reality, this appearance results from a sequential relationship between the images.

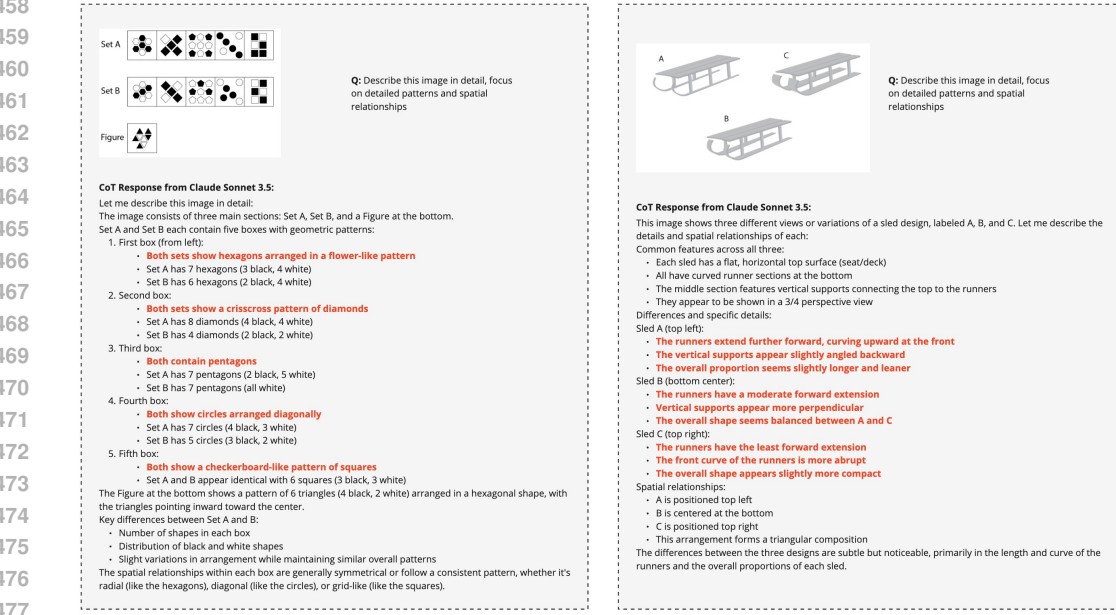

Figure 12: Another example of Claude 3.5 Sonnet's failure to capture detailed visual logic is its hallucination of spatial facts about the relationship between the triangle and the circle. Claude incorrectly assumes that the circle is splitting in half, when in reality, this appearance results from a sequential relationship between the images.

# G  MCQ vs CoT Evaluations

We found that when MLLMs are prompted to produce CoT reasoning, they often provide incorrect reasoning, leading to lower scores, as open-ended evaluations account for the quality of the reasoning itself.

According to the benchmark outcome, we show that MCQ-based evaluations consistently result in higher raw scores compared to CoT-based evaluations in almost all categories for reasoning and capabilities. This mirrors human behavior, where it is often easier to make an educated guess and get the answer right without fully understanding the question or providing correct reasoning. Since MCQ evaluations may overlook cases where MLLMs guess the answer correctly without valid reasoning, we argue that CoT evaluations offer a more reliable measure of MLLM reasoning capabilities, as they assess both the answer and the reasoning behind it.

# H  Model Size and Performance

We also observe that performance on LogicVista generally increases with model parameter sizes. As illustrated in Figure 13, there is a positive correlation between model size and average LogicVista performance. This trend suggests that larger models may possess greater capacities for learning and understanding complex relationships, allowing them to better tackle the demands of visual logical reasoning tasks. This improvement may be attributed to their ability to capture more intricate patterns and nuances in data, which enhances their overall reasoning capabilities. However, it is important to note that while larger models tend to perform better, this does not guarantee that all larger models will excel equally, as other factors such as training data quality and model architecture also play significant roles in determining performance.

# I  Examples of LogicVista Logical Reasoning Data

Table 6: Three samples requiring inductive logical reasoning skills.

**(a)**

| | |
|---|---|
| **Q:** | Which choice (A, B, C, or D) completes the series? |
| **Answer:** | D |
| **Reasoning:** | In this example, there are two rules to be applied. The first is that the circle moves counter-clockwise in the hexagon. It follows that, in the following diagram, the circle will be in the upper corner of the hexagon, pointing to D as the answer. To confirm this, the second rule can be applied, according to which the position of the black triangle alternates between bottom left and top right. Thus, in the following diagram, the black triangle will need to be in the upper right corner of the hex. The answer is therefore definitely D. |
| **Logical Reasoning Skill:** | Inductive |
| **Required capability** | Diagram |

**(b)**

| | |
|---|---|
| **Q:** | Who is the odd-one-out? Select answers from A-I. |
| **Answer:** | G |
| **Reasoning:** | Element G constitutes the exception and is therefore the correct answer. |
| **Logical Reasoning Skill:** | Inductive |
| **Required capability** | Diagram |

**(c)**

| | |
|---|---|
| **Q:** | Two grids containing colored symbols and following a common rule are presented. In the block on the right, four additional grids are presented. The candidate must find the two grids that follow the same rule out of these four options. What options (A, B, C, or D) follow this same rule? |
| **Answer:** | B, D |
| **Reasoning:** | In this example, it is easy to see that the rule governing the two grids on the left is: blue triangles are present in each of the two bottom lines. This rule is followed in the two grids on the right. |
| **Logical Reasoning Skill:** | Inductive |
| **Required capability** | Diagram, OCR |

Table 7: Three samples requiring deductive logical reasoning skills.

---

**(a)**

*All footballers are fit and healthy.*

*All famous sports players are footballers.*

*Given that the above is true, which of the following is the logical deduction?*

1. *All footballers are famous sports people*

2. *All famous people are fit and healthy*

3. *All famous sports players are fit and healthy*

4. *All fit and healthy people are footballers*

5. *All football players are men*

| | |
|---|---|
| **Q:** | Which is the correct answer according to the image? Select from 1-5? |
| **Answer:** | 3 |
| **Reasoning:** | Using deductive reasoning, the only logical answer is 3. To get to this answer, you need to simplify the given facts. All famous sports players are footballers, and all footballers are fit and healthy. We can not deduce that all footballers are famous sports people, as we have not got that information. We can not deduce that all famous people are fit and healthy, because the fact is about famous sports people. This is the logical answer. This information is not given; all footballers are fit and healthy but we can not logically link that all fit and healthy people are footballers. This is obviously incorrect, as gender is not mentioned at all in the question. |
| **Logical Reasoning Skill:** | Deductive |
| **Required capability:** | OCR |

---

**(b)**

The vast majority of swallows are blue. What is the most logical conclusion?

A. There is a white swallow.

B. Not everything that is blue is a swallow.

C. There is a blue swallow.

D. None of the answers are satisfactory.

| | |
|---|---|
| **Q:** | What is the correct answer to the question in the image? Select from A-D? |
| **Answer:** | C |
| **Reasoning:** | The vast majority of swallows are blue so the answer must be C: there is a blue swallow. |
| **Logical Reasoning Skill:** | Deductive |
| **Required capability:** | OCR |

---

**(c)**

> The people determine what is produced.
> The government is made up of the people.
> Production is determined by the free-market.
> The free-market is made up of production.
> Government is determined by the free-market.

| | |
|---|---|
| **Q:** | What is produced is determined by the people. Select from A, B and C. (A) True (B)False (C)Insufficient Information? |
| **Answer:** | A |
| **Reasoning:** | Line 1 states that the people determine what is produced. Line 2 states that the government is made up of the people. Therefore, the people determine what is produced. This is a syllogism. Thus, this statement is true. |
| **Logical Reasoning Skill:** | Deductive |
| **Required capability:** | OCR |

---

Table 8: Three samples requiring numerical logical reasoning skills.

**(a)**

**Share Price Index**

| Company | Today's Price (€) | Change from previous day (%) | Past 12 months | |
|---|---|---|---|---|
| | | | Max price (€) | Min price (€) |
| Huver Co. | 1,150 | 1.10 | 1,360 | 860 |
| Drebs Ltd | 18 | 0.50 | 22 | 11 |
| Fevs Plc | 1,586 | -9.00 | 1,955 | 1,242 |
| Fauvers | 507 | -1.00 | 724 | 464 |
| Steapars | 2,537 | 1.00 | 2,630 | 2,216 |

**Dividend Index**

| Dividend paid per share (€) | Huver Co. | Drebs Ltd | Fevs Plc | Fauvers | Steapars |
|---|---|---|---|---|---|
| Interim Dividend | 0.83 | 0.44 | 0.34 | 0.09 | 0.48 |
| Final Dividend | 1.75 | 1.12 | 1.25 | 0.32 | 0.96 |

**Note:** the total annual dividend paid per share is the sum of the interim dividend and the final dividend.

**Q:** Which share had the largest difference between highest and lowest price over the last 12 months? Select from A, B, C, D and E. (A) Huver Co. (B) Drebs Ltd (C) Fevs Plc (D) Fauvers (E) Steapars

**Answer:** C

**Reasoning:** Step 1- Calculate the difference between the maximum and the minimum prices. Huver Co. = 1,360 - 860 = 500 Drebs Ltd = 22 - 11 = 11 Fevs Plc = 1,955 - 1,242 = 713 Fauvers = 724 - 464 = 260 Steapars = 2,630 - 2,216 = 414. Tip: Notice the wording of the question is asking for the share with the largest absolute change in price, NOT the largest percentage change, which would have been Drebs Ltd. If the question had wanted the percentage change it would have used the word percentage. Thus the correct answer is (C) Fevs Plc

**Logical Reasoning Skill:** Numerical

**Required capability:** OCR

**(b)**

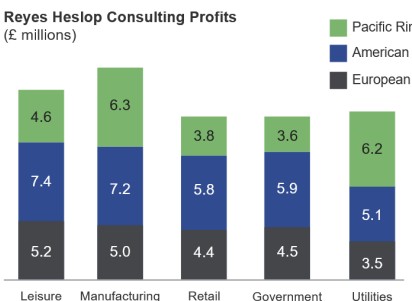

**Q:** Reyes Heslop had a target for Leisure profits to be a quarter of their total profits. Assuming profits in other areas remain the same, by how much did the Leisure profits miss this target? Select from A, B, C, D and E. (A) 31.8 million (B) 32.4 million (C) 32.7 million (D) 33.2 million (E) 33.4 million

**Answer:** D

**Reasoning:** Step 1- Calculate the total Reyes Heslop profits across all areas other than Leisure. (6.3 + 7.2 +5.0) + (3.8 + 5.8 + 4.4) + (3.6 + 5.9 + 4.5) + (6.2 +5.1 + 3.5) = 61.3million. Step 2- This needs to be / of all profits for the condition to be met. Therefore all profits, across all sectors, would be 61.3 / 75% = 81.7333million. Step 3- Now we look at the difference between actual and target Leisure profits. Actual = (4.6 + 7.4 + 5.2) = 17.2 Target = (81.7333 - 61.3) = 20.4333 Shortfall = 3.2333 (millions) Thus the correct answer is (D) 33.2million

**Logical Reasoning Skill:** Numerical

**Required capability:** Diagram, OCR

**(c)**

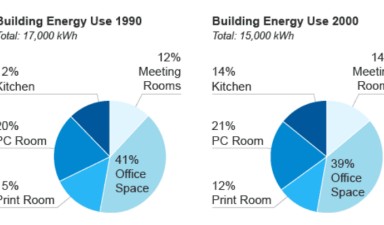

**Q:** Which space experienced the smallest reduction in kWh used between 1990 and 2000? Select from A, B, C, and D. (A) Office Space (B) Print Room (C) Meeting Rooms (D) PC Room

**Answer:** D

**Reasoning:** Step 1- Calculate the value of kWh for 1990 and 2000 for each of the rooms. Room 1990 per kWh 2000 per kWh Meeting Rooms 2.04 2.10 Office Space 6.97 5.85 Print Room 2.55 1.80 PC Room 3.40 3.15 Kitchen 2.04 2.10 Step 2- Subtract the kWh for 2000 from that of 1990 for each of the rooms. Room change (1990 - 2000) kWh Meeting Rooms -0.06 Office Space 1.12 Print Room 0.75 PC Room 0.25 Kitchen -0.06 Step 3- Look for the smallest positive value. Negative values represent an increase between 1990 and 2000. Tip- You only need to perform 4 calculations, as two of the rooms have the same values. Thus, the correct answer is (D) PC Room.

**Logical Reasoning Skill:** Deductive

**Required capability:** Diagram, OCR

Table 9: Three samples requiring spatial logical reasoning skills.

**(a)**

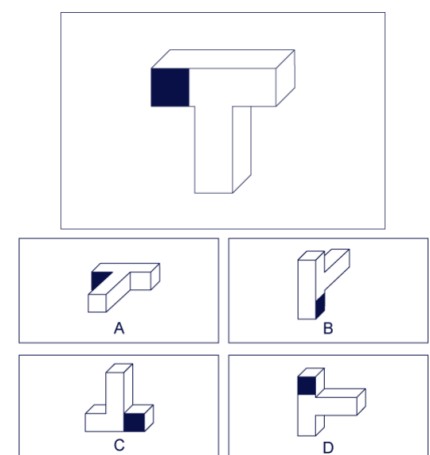

| | |
|---|---|
| **Q:** | Which figure is a rotation of the object? Select from A, B, C, and D. (A) (B) (C) (D) |
| **Answer:** | B |
| **Reasoning:** | The answer is B. |
| **Logical Reasoning Skill:** | Spatial |
| **Required capability:** | Diagram |

**(b)**

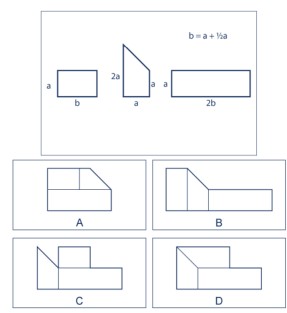

| | |
|---|---|
| **Q:** | Which figure can be formed with the given piece? Select from A, B, C, and D. (A) (B) (C) (D) |
| **Answer:** | C |
| **Reasoning:** | The answer is C. |
| **Logical Reasoning Skill:** | Spatial |
| **Required capability:** | Diagram |

**(c)**

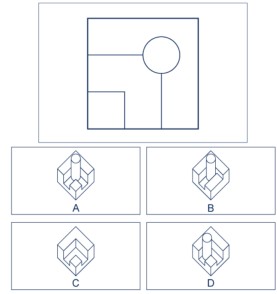

| | |
|---|---|
| **Q:** | To which object does the given top view correspond? Select from A, B, C, and D. (A) (B) (C) (D) |
| **Answer:** | A |
| **Reasoning:** | The answer is A. |
| **Logical Reasoning Skill:** | Spatial |
| **Required capability:** | Diagram |

Table 10: Three samples requiring mechanical logical reasoning skills.

**(a)**

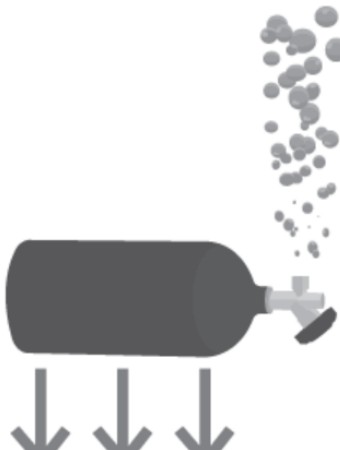

| | |
|---|---|
| **Q:** | A non-pressurised cylindrical metal tank filled with air is submerged underwater. As the air escapes, the tank gradually moves deeper underwater. Which statement provides the best reason for this motion? Select from A, B, C, D, and E. (A) The bubbles provide a downward thrust on the tank (B) The metal increases in density so it gets heavier (C) The bubbles lower the density of the water which lowers its buoyancy (D) Water replaces the air in the tank which makes it heavier (E) Impossible to tell |
| **Answer:** | D |
| **Reasoning:** | As air escapes the available space is quickly replaced with water, so the tank's density becomes the same as that of the water and with the added weight and density of the tank itself continues to sink. |
| **Logical Reasoning Skill:** | Mechanical |
| **Required capability:** | Diagram |

**(b)**

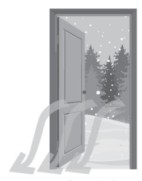

| | |
|---|---|
| **Q:** | It is a cold winter outside and a well-insulated house has its heater turned on. The front door is opened and cold air rushes in. If the wind speed outside is very low, how would the cold air enter the house? Select from A, B, C, D, and E. (A) Scenario A, the cold air will flow towards the floor (B) Scenario B, the cold air will flow towards the ceiling (C) A combination of A and B (D) The cold air will not enter the house (E) Impossible to tell |
| **Answer:** | A |
| **Reasoning:** | Cold air sinks, whereas hot air rises. The house and the air inside it are warmer than the outside air temperature, so if these two systems (house and outside) were to be suddenly connected (door opening) the cold air would sink and the hot air would sit above the cold air until the heat transferred between the two. |
| **Logical Reasoning Skill:** | Mechanical |
| **Required capability:** | Diagram |

**(c)**

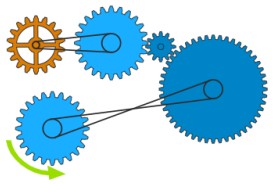

| | |
|---|---|
| **Q:** | In which direction does the orange gear rotate? Select from A, B, and C. (A) Clockwise (B) Counterclockwise (C) No rotation |
| **Answer:** | A |
| **Reasoning:** | The correct answer is clockwise. |
| **Logical Reasoning Skill:** | Mechanical |
| **Required capability:** | Diagram |

1782
1783
1784
1785
1786
1787
1788
1789
1790
1791
1792
1793
1794
1795
1796
1797
1798
1799
1800
1801
1802
1803
1804
1805
1806
1807
1808
1809
1810
1811
1812
1813
1814
1815
1816

**Model Size vs Average Reasoning and Capability Accuracy**

Figure 13: Correlation between Model Size and Average Accuracy. The scatter plot employs varying dot sizes to indicate the number of models with identical model sizes, illustrating the distribution density.

1817
1818
1819
1820
1821
1822
1823
1824
1825
1826
1827
1828
1829
1830
1831
1832
1833
1834
1835

## J    EXAMPLES OF DIFFERENT BROAD LOGICVISTA CAPABILITIES DATA

Table 11: Three samples of diagram, OCR, and mixed LogicVista data

---

**(a)**

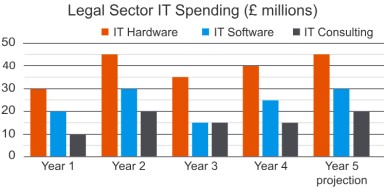

| | |
|---|---|
| **Q:** | Which ball is the heavies? Select from A, B, C, and D. (A) A (B) B (C) C (D) CAN NOT SAY |
| **Answer:** | D |
| **Reasoning:** | The correct answer is D. |
| **Logical Reasoning Skill:** | Mechanical |
| **Required capability:** | Diagram |

---

**(b)**

Which of these objects will not float on water?

| | |
|---|---|
| **Q:** | Select from A, B, C, and D. (A) banana (B) scissors (C) empty plastic soda bottle (D) wooden pencil |
| **Answer:** | B |
| **Reasoning:** | The correct answer is B because scissors have metal and are most likely to sink. |
| **Logical Reasoning Skill:** | Deductive |
| **Required capability:** | OCR |

---

**(c)**

Legal Sector IT Spending (£ millions)

Two Legal Sector IT Firms Income for Consultancy Services (10,000s)

| | Make Fit Ltd | Pure Gap Plc |
|---|---|---|
| Year 1 | 290 | 230 |
| Year 2 | 180 | 310 |
| Year 3 | 260 | 300 |
| Year 4 | 320 | 290 |

| | |
|---|---|
| **Q:** | Which of the following statements is false regarding legal sector spending between Year 4 and projected Year 5? Select from A, B, C, D and E. (A) IT consulting will increase by 35million. (B) IT consulting will match that of year 2. (C) IT software will exceed IT consulting. (D) Spending on IT hardware will decline. (E) None of these. |
| **Answer:** | D |
| **Reasoning:** | Step 1- Check in turn whether each statement is true or false: a) The projected spend on IT consulting is projected to increase by 35 million. Option A is true. b) The projected spend on IT consulting is 320 million, which matches year 2. Option B is true. c) The projected spend on IT software is 330 million and for IT consulting it is 320 million. Option C is true. d) There are increases projected for IT hardware, for IT software and for consulting, therfore s̈pending on IT hardware will declineïs not true. The option for D is false. e) We see that option D is false, so E cannot be the correct answer. Thus the correct answer is (D) Spending on IT hardware, software and consulting is projected to decline. |
| **Logical Reasoning Skill:** | Numerical |
| **Required capability:** | Diagram, OCR |

---

## K    DATA LEAKAGE CONCERNS OF EXISTING BENCHMARKS

As shown in 14, sourcing data from gated sources allows LogicVista to greatly minimize the risk of data leakage. In an experiment, we randomly sampled 50 images from datasets such as MM-vet,

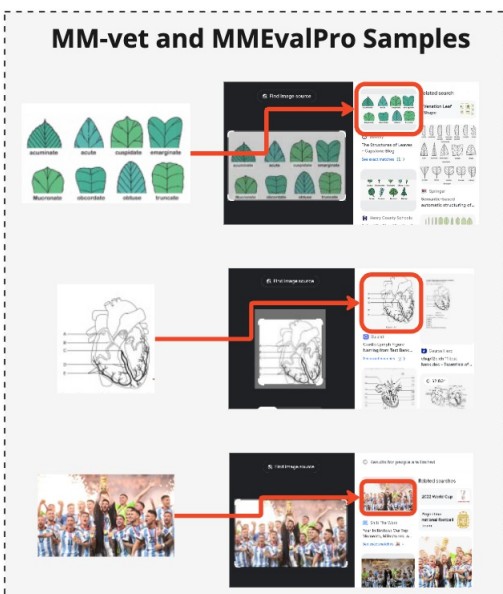 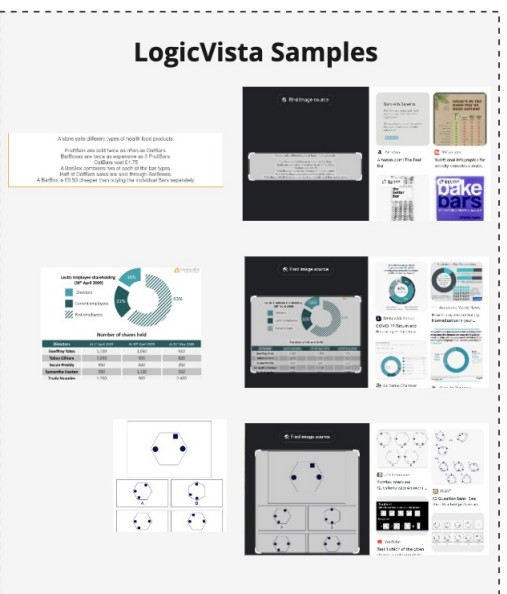

Figure 14: LogicVista mitigates potential data leakage by sourcing from gated private datasets (with permission). This approach ensures a fair comparison by isolating MLLM reasoning abilities, preventing any overlap with information that may have been included in their training data.

MMEvalPro, RAVEN, and MathVista Yu et al. (2023); Huang et al. (2024); Zhang et al. (2019); Lu et al. (2024), and used Google's reverse image search. We found that all samples from existing benchmarks were publicly available online, whereas nearly all of LogicVista's samples were inaccessible, either behind paywalls or requiring registration. Since most of LogicVista's data is not publicly available, it is much more difficult to scrape for training MLLM models. This restricted access reduces the chances of LogicVista's samples being included in training datasets, unlike in open benchmarks.

## L    CROWDSOURCING ANNOTATION TOOL

To scale LogicVista for the future, we have released an annotation tool similar to the one used in our annotation process. This tool facilitates robust annotations by incorporating rounds of peer review before finalizing entries in LogicVista. Additionally, it is web-based, allowing the community to contribute to LogicVista from anywhere. We hope this will enable LogicVista to grow and increase its sample size significantly.

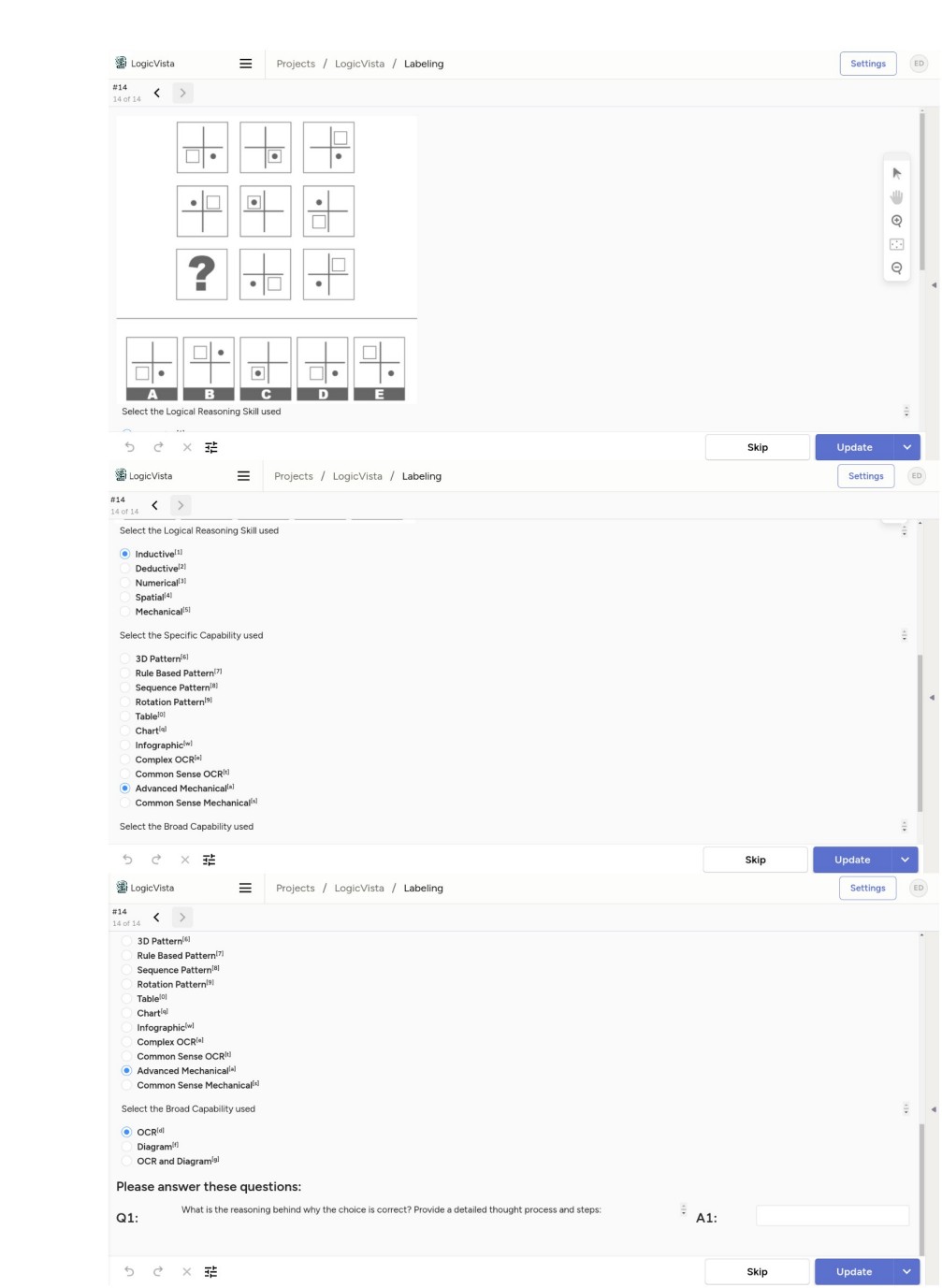

Figure 15: Example of the annotation process using our tool, enabling the community to contribute to scaling LogicVista effectively.

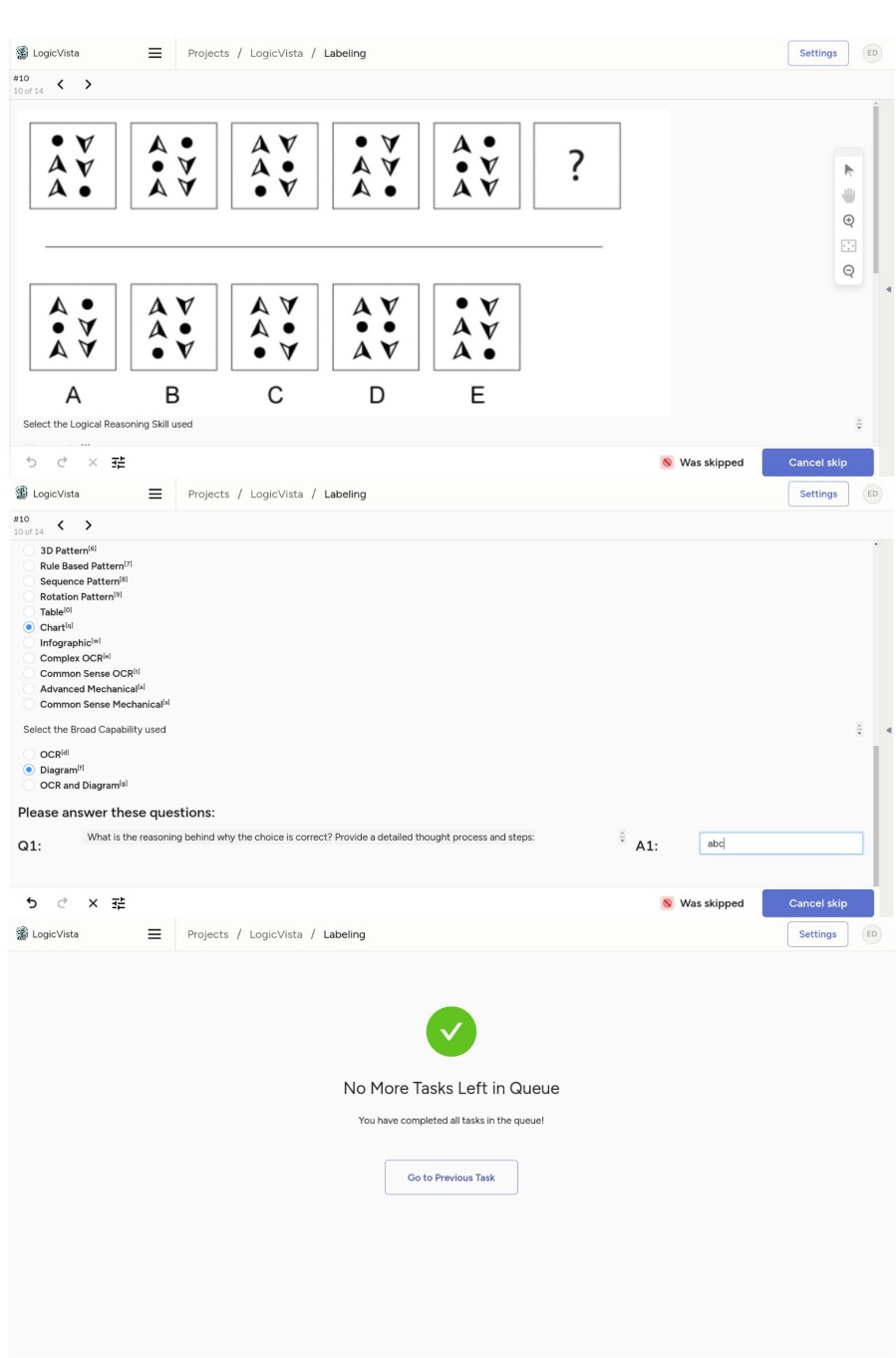

Figure 16: Additional example of annotation process using our crowdsourcing tool

