# OpenReview forum: "LogicVista: Multimodal LLM Logical Reasoning Benchmark in Visual Contexts"
_ICLR.cc/2025/Conference — ICLR 2025 Conference Withdrawn Submission_

### Official Review · Reviewer_pxHL · 2024-10-19

**Soundness:** 2
**Presentation:** 2
**Contribution:** 3
**Rating:** 5
**Confidence:** 4

**Summary:**

This paper proposes an evaluation benchmark named LogicVista, which is designed to assess the logical reasoning abilities of Multimodal Large Language Models (MLLMs) in visual contexts. LogicVista comprehensively evaluates 11 existing MLLMs through five types of logical reasoning tasks. These tasks cover inductive, deductive, numerical, spatial, and mechanical reasoning, using 448 multiple-choice questions (MCQs) with correct answers and human-written reasoning annotations. The evaluation methods include both MCQ and open-ended Chain-of-Thought (CoT) analysis to better understand the models' strengths and limitations. Experimental results show that while some models perform well in deductive reasoning, most models score lower in other reasoning categories.

**Strengths:**

1. The paper introduces a new evaluation benchmark, LogicVista, focusing on evaluating visual logical reasoning abilities.
2. The structure of the paper is clear, and it thoroughly explains the design motivation, data collection methods, evaluation models, and result analysis of LogicVista.

**Weaknesses:**

1. Although LogicVista covers various logical reasoning tasks, the sample size of 448 may be insufficient to fully capture the performance of MLLMs in real-world complex scenarios.
2. The tasks in LogicVista mainly focus on basic logical reasoning, such as mechanical and inductive reasoning. There is a lack of sufficient coverage for higher-level complex reasoning tasks, such as multi-step reasoning or continuous multimodal reasoning.
3. The paper mentions the risk of data leakage in many benchmarks, indicating that MLLMs might have encountered some test data during training. Although LogicVista avoids publicly available internet data in its dataset selection, it does not provide detailed mechanisms to verify the complete independence of all samples.

**Questions:**

Please refer to the Weaknesses.

---

### Official Review · Reviewer_LqnE · 2024-11-01

**Soundness:** 3
**Presentation:** 3
**Contribution:** 3
**Rating:** 5
**Confidence:** 4

**Summary:**

This paper presents a new benchmark, LogicVista, to assess MLLMs' logical reasoning capabilities, addressing an overlooked area in AI evaluation. It includes 448 annotated tasks across five reasoning types (inductive, deductive, numerical, spatial, mechanical) and nice task categories, providing a systematic test of MLLMs' logical reasoning.

**Strengths:**

- Originality: provides a new, high-quality dataset
- Quality: strict measures are taken to prevent data leakage; well-designed experiments
- Clarity: clear data collection and experimental procedures.
- Significance: compensates for the lack of benchmarks specifically designed to test logical reasoning.

**Weaknesses:**

Relatively small dataset — 448 samples

In my assessment, a high-quality benchmark typically excels in at least one of three areas:
1. **Data Quality**: The benchmark offers exceptionally high-quality data that often serves as an evaluation standard within its field, as exemplified by VQAv2 and MMMU.
2. **Tooling and Resources**: It provides innovative tools or resources, such as novel code architectures, metrics, or other practical assets that advance usability and applicability.
3. **Research Insight**: The benchmark highlights a previously overlooked problem or dimension, encouraging new avenues for research and fostering deeper understanding.

While the quality of this work is solid, it falls short of providing substantial new insights or advancements in these areas. As a result, its impact on the field may be limited.

**Questions:**

**Major Points:**

- Is there theoretical justification for the representativeness of the five selected reasoning task categories?
- Would it be beneficial to include an error analysis section?
- Uncertain if the significance of this work aligns with the claims made.

**Minor Comments:**

- In line 50, is GLoRE applied to an MLLM or just an LLM?
- When "MCQ-based" and "CoT-based" are first mentioned, it may be clearer to use the full terms, such as "Chain of Thought-based (CoT-based)."

---

### Official Review · Reviewer_7rfs · 2024-11-03

**Soundness:** 2
**Presentation:** 3
**Contribution:** 2
**Rating:** 6
**Confidence:** 4

**Summary:**

This paper presents a variety of logical reasoning tasks, allowing for a comprehensive assessment of the model’s performance across different logical contexts.

**Strengths:**

1. This approach covers a variety of logical reasoning tasks, allowing for a comprehensive assessment of the model’s performance across different logical contexts.
2. The data sources are drawn from authorized intelligence tests, which effectively ensures data privacy and novelty, reducing the risk of training data leakage.
3. The evaluation methods are rich, combining multiple-choice (MCQ) and chain-of-thought (CoT) approaches, adding depth to the assessment. This allows for effective evaluation of the model’s reasoning process as well as precise measurement of answer accuracy.

**Weaknesses:**

1. LogicVista presents significant difficulty differences across reasoning tasks, leading to unbalanced performance. For example, while models perform relatively well in deductive and mechanical reasoning, they perform less effectively in inductive, numerical, and spatial reasoning tasks. This variation may affect the fairness of overall performance evaluations.
2. The paper notes that current visual encoders face substantial limitations in recognizing spatial and abstract relationships, particularly in complex spatial reasoning and 3D pattern recognition tasks, where models tend to underperform. However, the paper lacks specific improvement suggestions, such as ways to enhance visual encoders or improve training data to boost reasoning ability, which somewhat constrains future developmental direction.
3. There is a lack of ablation studies to deeply analyze the specific contributions of each component to overall performance. Ablation studies can reveal the independent impact of each module, helping to better understand model composition and optimization paths.

**Questions:**

I would like to ask the authors about the weakness and how they can improve them.

---

### Official Review · Reviewer_BAas · 2024-11-04

**Soundness:** 3
**Presentation:** 3
**Contribution:** 2
**Rating:** 5
**Confidence:** 4

**Summary:**

The paper proposes the LogicVista benchmark dataset, aimed at systematically evaluating the performance of multimodal large language models in visual reasoning tasks. The data is sourced from authorized intelligence tests and employs a dual evaluation method combining multiple-choice questions (MCQ) and chain-of-thought (CoT). It covers inductive, deductive, numerical, spatial, and mechanical reasoning. The experiments reveal an imbalance in model performance across different reasoning tasks, with particular limitations in complex spatial reasoning tasks.

**Strengths:**

1. Innovative Multimodal Reasoning Evaluation: The paper systematically evaluates the logical reasoning capabilities of multimodal large language models (MLLMs), covering five core areas: deductive reasoning, inductive reasoning, numerical reasoning, spatial reasoning, and mechanical reasoning. This evaluation fills gaps in current methodologies and holds significant potential for advancing research in the multimodal domain.
2. Rigorous Data Source: The paper uses authorized IQ test data, which avoids the common issue of public data leakage and ensures fairness in the evaluation. This data is more reflective of the models’ reasoning capabilities, rather than merely testing memory or simple inference abilities.
3. Multidimensional Evaluation Approach: By combining MCQ and Chain-of-Thought (CoT) methods, the paper efficiently quantifies the model’s selection ability while also deeply analyzing its reasoning process, balancing both evaluation depth and efficiency.
4. Scalability and Continuous Updates: The introduction of crowdsourced annotation tools ensures the dataset’s scalability and future updates, laying a solid foundation for iterative evaluations.
5. Clear Language and Logical Presentation: The paper is clearly written, with well-structured logic and organized presentation of experimental results, aiding readers in understanding its contributions.

**Weaknesses:**

1. Limited Data Source: Although the IQ test data is rigorous, it is relatively concentrated in scope, lacking coverage of other fields (e.g., scientific reasoning, language understanding). The dataset size is also limited, potentially hindering the ability to fully capture the model’s performance across a broader range of tasks.
2. Lack of Depth in Benchmark Design and Ablation Studies: The paper primarily uses random and frequentist baselines, which provide limited insights into the deeper aspects of reasoning. The absence of ablation studies restricts the analysis of how different model components contribute to performance. Introducing more complex baselines and conducting ablation studies could enhance the interpretability of the results.
3. Redundant Logical Expressions: The distinction between the “Broad Capabilities” and “Specific Capabilities” sections in 4.2 is somewhat redundant, particularly in the discussion of OCR and diagram tasks. Simplifying these sections and clearly distinguishing between the two could improve the clarity of the paper’s logic.

**Questions:**

I don't have questions.

---

### Official Review · Reviewer_HBcL · 2024-11-04

**Soundness:** 2
**Presentation:** 3
**Contribution:** 2
**Rating:** 3
**Confidence:** 4

**Summary:**

This paper presents LogicVista, a benchmark for evaluating MLLMs' logical reasoning abilities. It includes 448 multi-choice questions spanning 11 abilities, with human-annotated rationales. Experiments on 11 MLLMs show there remains a huge gap for improvement.

**Strengths:**

- A newly proposed human-annotated benchmark for MLLMs' logical reasoning abilities, sourced from gated private datasets.
- The dataset is indeed challenging considering the low scores of Claude and GPT-4o, with extensive analysis on cases and different components.

**Weaknesses:**

Although I believe it will be a valuable data resource (if the authors agree to open-source), my main concern is the **necessity** of this dataset in these aspects:

(1) There are already many datasets in the field of reasoning, such as MathVista [1], MMMU [2] and ScienceQA [3], with **some subsets even overlapping with LogicVista** (such as IQTest in MathVista).

(2) As the dataset is sourced from 15 private IQ tests, why are **IQ tests** used in this dataset specifically designed for logical reasoning? Need some citations to support this claim.

(3) The **categorisation** of skills, broad and specific capabilities need further explanations or authorized references. Did you refer to previous MLLM evaluation research? For example, FLASK [4] refers to this QA taxonomy [5] for the definition of skills.

(4) Regarding the **quantity** of this dataset, there are only 448 questions, much smaller than datasets in this category, such as ScienceQA (21208 questions) and MathVista (6141 questions).

[1] MathVista: Evaluating Mathematical Reasoning of Foundation Models in Visual Contexts https://arxiv.org/abs/2310.02255

[2] MMMU: A Massive Multi-discipline Multimodal Understanding and Reasoning Benchmark for Expert AGI https://arxiv.org/abs/2311.16502

[3] Learn to Explain: Multimodal Reasoning via Thought Chains for Science Question Answering https://scienceqa.github.io/

[4] FLASK: Fine-grained Language Model Evaluation based on Alignment Skill Sets https://arxiv.org/abs/2307.10928

[5] QA Dataset Explosion: A Taxonomy of NLP Resources for Question Answering and Reading Comprehension https://arxiv.org/abs/2107.12708

**Some minor weaknesses:**

- Lack of human annotation details: As the authors state there is cross-validation during annotation, what is the inter-annotator agreement? How much payment is given to each annotator? How many hours does it take to finish annotation?

- Reliability of LLM-based evaluator on the chain-of-thought: What is the agreement rate between human-based and LLM-based evaluators on the rationales? Only when the agreement is acceptable can the LLM-based evaluation be convincing.

**Questions:**

- The abbreviation of MCQ needs to be clarified the first time using it.
- The pie chart in the middle of Figure 3 mixes with the right chart.

---

### Note · Authors · 2024-11-15

**Comment:**

We thank the reviewers for their thoughtful feedback, constructive comments, and helpful suggestions. We have decided to withdraw our submission to refine further and enhance our work for future resubmission.

**Withdrawal Confirmation:**

I have read and agree with the venue's withdrawal policy on behalf of myself and my co-authors.